# Investigating ENSO and its teleconnections under climate change in an ensemble view – a new perspective

Tímea Haszpra[1,2,†], Mátyás Herein[1,2,†], and Tamás Bódai[3,4]

[1]Institute for Theoretical Physics, Eötvös Loránd University, Budapest, Hungary
[2]MTA–ELTE Theoretical Physics Research Group, Eötvös Loránd University, Budapest, Hungary
[3]Pusan National University, Busan, Republic of, Korea
[4]Center for Climate Physics, Institute for Basic Science, Busan, Republic of Korea
[†]These authors contributed equally to this work.

**Correspondence:** hatimi@caesar.elte.hu

**Abstract.** The changes in the El Niño–Southern Oscillation (ENSO) phenomenon and its precipitation-related teleconnections over the Globe under climate change are investigated in the Community Earth System Model's Large Ensemble from 1950 to 2100. For the investigation, a recently developed ensemble-based method, the snapshot empirical orthogonal function (SEOF) analysis is used. The instantaneous ENSO pattern is defined as the leading mode of the SEOF analysis carried out at a given time instant over the ensemble. The corresponding principal components (PC1s) characterize the ENSO phases. Considering SST regression maps, we find that the largest changes in the typical amplitude of sea surface temperature fluctuations occur in the June–July–August–September (JJAS) season, in the Niño3–Niño3.4 region and in the western part of the Pacific Ocean, however, the increase is also considerable along the Equator in December–January–February (DJF). The Niño3 amplitude shows also an increase of about 20% and 10% in JJAS and DJF, respectively. The strength of the precipitation-related teleconnections of the ENSO is found to be non-stationary, as well. For example, the anti-correlation with precipitation in Australia in JJAS and the positive correlation in Central and North Africa in DJF are predicted to be more pronounced by the end of the 21th century. Half-year-lagged correlations, aiming to predict precipitation conditions from ENSO phases, are also studied. The Australian, Indonesian precipitation and that of the eastern part of Africa in both JJAS and DJF seem to be well predictable based on ENSO phase, while the South Indian precipitation is in relation with the half-year previous ENSO phase only in DJF. The strength of these connections increases, especially from the African region to the Arabian Peninsula.

## 1 Introduction

The El Niño—Southern Oscillation (ENSO) is recognized as the dominant interannual fluctuation in the climate system (see, e.g., Bjerknes, 1969; Rasmusson and Carpenter, 1982; Neelin et al., 1998; Philander, 1990; Timmermann et al., 2018). This naturally occurring fluctuation originates in the tropical Pacific region and affects weather and climate worldwide. The warm (cold) phase of the ENSO, called El Niño (La Niña), is associated with above (below) average sea surface temperature in the central and eastern equatorial part of the Pacific Ocean. The ENSO cycle also has several regional impacts on precipitation and temperature over the Globe. For example, during El Niño episodes Australia, Indonesia in both December–February and

June–September, and India and the equatorial band in Africa in December–February experience a reduced amount of rainfall, while Peru and Chile has wetter than normal weather in July–September (see, e.g., Diaz et al. (2001); Yang and DelSole (2012); Yeh et al. (2018)).

Therefore, if the ENSO changes it may have large climatic impacts (Trenberth et al., 1998; Wallace et al., 1998; Glantz et al., 2001; Trenberth et al., 2002; Guilyardi et al., 2009; Collins et al., 2010; Vecchi and Wittenberg, 2010; Cai et al., 2015). Consequently, an open and crucial question is how ENSO will change in the changing climate. There has been many studies aiming to answer this question, however, the model simulations of future ENSO changes diverge widely among climate models (Yeh and Kirtman, 2007; Stevenson, 2012; Christensen et al., 2013; Bellenger et al., 2014).

Most of the studies agree on using temporal statistics (including means, variances, correlations, etc.) applied to a time-dependent dynamical system, i.e., in our changing climate. However, a correct application of temporal statistics requires stationarity, as argued in Drótos et al. (2015, 2016), which does not hold in a changing climate. The non-stationarity of the climate system naturally appears in the simulations generated for climate projections by general circulation models, and it is especially important in the analysis of teleconnections (Herein et al., 2016, 2017; Roy et al., 2019; Chung et al., 2019).

To avoid the above-mentioned discrepancy of temporal methods, in this study we present an ensemble-based analysis. In this approach the relevant quantities of the climate system are the statistics taken at any given time instant over an ensemble of possible climate realizations. We note that a "time instant" can also mean time averages over certain periods, because the snapshot framework is also applicable for quantities evaluated over time intervals (Drótos et al., 2015). These ensembles typically evolve from slightly different initial conditions. In the context of climate, this kind of ensembles was used, e.g., for the investigation of the forced response and associated uncertainties arising from internal variability (Deser et al., 2012; Daron and Stainforth, 2013; Kay et al., 2015; Suarez-Gutierrez et al., 2018). They are also utilized on specific topics, e.g., for the estimation on how many ensemble members are needed to detect a statistically significant strengthening of the Northern Hemisphere polar vortex due to volcanic eruptions (Bittner et al., 2016), to reveal that the more rapidly warming surface in climate models compared to observations can be caused by internal variability in the top-of-atmosphere energy imbalance (Hedemann et al., 2017), for unfolding that internal variability of the ocean carbon uptake is as large as the forced temporal variability (Li and Ilyina, 2018), and the internal variability has a dominant role in the strengthening of the Pacific Walker circulation (Chung et al., 2019). Large ensembles prove to be useful also in revealing the deviations and their reasons between the results of traditional single-time series statistics and ensemble statistics (Herein et al., 2016), and in studying the NAO teleconnections (Herein et al., 2017). Large ensemble simulations were also investigated in low-dimensional systems (see, e.g. Bódai et al., 2011; Bódai and Tél, 2012; Drótos et al., 2015). The mathematical concept that provides the appropriate framework is that of snapshot (Romeiras et al., 1990; Drótos et al., 2015) or pullback attractors (Arnold, 1998; Ghil et al., 2008; Chekroun et al., 2011). The applicability of this framework was also established by laboratory experiments (Vincze et al., 2017). The application of ensembles in the snapshot framework is overviewed in Tél et al. (2019).

In the context of climate simulations, this framework implies that a climate simulation with any initial condition after a transient time (during which it forgets its initial condition) converges to the snapshot attractor which describes the "permitted" climate states under the external forcing history up to that time, such as $CO_2$ concentration, etc. Furthermore, starting

a sufficiently large ensemble of climate simulations with slightly different initial conditions is proved to correctly cover the distribution of the possible climate states of the snapshot attractor after the transient time at any time instant, therefore, from

that time on the ensemble can be used to characterize the potential states at each time instant (Drótos et al., 2015; Herein et al., 2016; Drótos et al., 2017). This ensemble can be also called parallel climate realizations (Herein et al., 2017; Tél et al., 2019). We note that it is remarkable that Leith (1978) came up with a similar idea as early as in 1978, however, Leith's work has not spread widely in the climate community.

The snapshot framework, which can be applied numerically to large ensembles, also provides a mathematically correct
method to separate the effect of internal variability from the forced response under climate change (see, e.g. Drótos et al., 2015), via, e.g., the ensemble standard deviation and the ensemble mean, respectively. Naturally, due to the time-dependence of the forcing, i.e., when the climate changes, the ensemble also undergoes a change in time, and as a consequence, both the mean state (average values) and the internal variability of the climate changes with time.

The snapshot framework can be applied also to phenomena analyzed by the widely used empirical orthogonal function
(EOF) analysis. Here we use a recently developed approach called the snapshot EOF analysis (SEOF) to reveal the potential changes in the ENSO and its teleconnections. This method has been introduced originally in Haszpra et al. (2019). It computes instantaneous SEOF loading patterns over the ensemble members at any given time instant, rather than with respect to the time dimension of any single ensemble member. Hence it is also capable of monitoring the time-dependence of the SEOF pattern. We note that a similar method (called EOF-E) has also been developed recently in Maher et al. (2018), however, it differs
from our SEOF method as it computes the EOF field for a given year partially over the ensemble dimension but constructing the sampling set from the different ensemble members and also from the different monthly fields of the given year with the seasonal mean signal previously removed.

This study focuses on the time evolution of the ENSO pattern and ENSO amplitude under the changing climate. Another crucial question is how ENSO teleconnections will be modified in the same time. Many studies have addressed this question
and reported that the ENSO teleconnections may change under the changing climate (Krishna Kumar et al., 1999; Yeh and Kirtman, 2007; Davey et al., 2014; Ramu et al., 2018; Yeh et al., 2018). However, most of the studies use the before-mentioned temporal statistic approach to get the relevant correlation coefficients of teleconnections, which introduces some subjectivity due to the choice of the time-window over which statistics are taken. This subjectivity was investigated, e.g., in Herein et al. (2017), in which it has been demonstrated that the traditional evaluation of correlation coefficients, carried out via temporal
statistics, provides incorrect or misleading results. Thus we emphasize that it is important to evaluate correlation coefficients and any other statistics with respect to the ensemble using the snapshot framework.

The SEOF method, computing all relevant quantities at single time instants, via the computed principal components (PC1s) of the leading SEOF mode used as certain ENSO indices to characterize the ENSO phases, also allows us to investigate ENSO teleconnections based only on instantaneous ensemble statistics. Since this can be done at any time instant it also enables us to
monitor the temporal evolution of the strength of the teleconnection during a climate change.

We also compute instantaneous ensemble-based correlation coefficients to characterize the connection of the ENSO phases with precipitation over the Globe at each year, and investigate the trends over time in the obtained correlation coefficient maps.

Lagged correlations between the two quantities can also be studied this way, providing the possibility of predicting precipitation based on PC1. We focus on the December–January–February (DJF) and the June–July–August–September (JJAS) season (for details see Data and Methods). In this way, in contrast to Maher et al. (2018) who uses all monthly data from a year, also the seasonal differences in the phenomenon can be investigated. We note that the SEOF method or any similar technique (e.g. EOF-E) can be applied successfully, providing robust statistics, only for large ensembles. Here, we choose to investigate the ENSO phenomenon in the large ensemble of one of the state-of the-art climate models, in the Community Earth System Model Large Ensemble Project (CESM-LE) (Kay et al., 2015). We emphasise that to our knowledge, this is the first time when the SEOF analysis using SST data was utilized to reveal changes in ENSO teleconnections.

The paper is organized as follows. Sect. 2 provides a brief overview of CESM-LE data and the SEOF analysis. It also includes a discussion on the capability of snapshot frameworks in general compared to the traditional single time series-based temporal analysis, and the interpretation of the meaning of their results. Sect. 3 presents the ensemble-based sea surface temperature regression maps as ENSO patterns, the ENSO amplitude, and their changes over time due to climate change. The precipitation-related teleconnections of the ENSO phenomenon and the alterations in their strength are also discussed in the section. Sect. 4 summarizes the main results and conclusions of the work.

## 2 Data and Methods

### 2.1 Data

For our study we use the meteorological fields of the CESM-LE produced by the fully-coupled CESM1 used in CMIP5 (Kay et al., 2015). Between 1920 and 2005 the CESM-LE simulations follow the CMIP5 historical experimental design (Taylor et al., 2012; Lamarque et al., 2010), while after 2005 to 2100 they follow the RCP8.5 scenario (Van Vuuren et al., 2011). In the study we utilize sea surface temperature (SST) and total precipitation (PRECT) fields with a horizontal resolution of $1° \times 1°$ and $1.25° \times 0.942°$, respectively. Due to the small systematic difference between the members run at NCAR and at the Toronto supercomputer (CESM-LE, 2016) we utilize only the members from the NCAR simulations. Taking into consideration the convergence time of the simulations we only deal with data from 1950 on.

### 2.2 Studying ENSO and its teleconnections in the snapshot framework

Here, we study ENSO by evaluating the variability of the SST field over the ensemble members at each time instant in the Pacific using the SEOF method. It is based on the region of [30°S, 30°N]×[100°E, 70°W], which is also chosen in Maher et al. (2018) to their EOF-E analysis. To eliminate the distorting impact of the regular latitude–longitude grid in the EOF analysis, the SST fields are weighted by the square root of the cosine of the latitude following Thompson and Wallace (2000). We consider the instantaneous ensemble-based leading SEOF mode (by which we mean the normalized eigenvectors associated with the largest eigenvalue of the covariance matrix of the SST anomaly fields) as the ENSO loading pattern, and the phase of the phenomenon in each member as the corresponding principal component (PC1). With the sign of the SEOF patterns which

shall be used in Fig. 1 PC1> 0 (PC1< 0) corresponds to anomalously warm (cold) events associated with above (below) the ensemble average SST in the central and east-central equatorial Pacific Ocean. As in Thompson and Wallace (2000) not the normalized EOF patterns, rather the SST regression maps are shown computed by regressing the unweighted SST anomaly fields onto the standardized PC1 data. Therefore, the values appearing on the regression maps characterize typical amplitudes in the variability of the SST. The instantaneous strength of ENSO is computed as the ensemble standard deviation of the PC1s of the given time instant as the snapshot counterpart of the temporal standard deviation of the PC1 used as a common practice to represent the strength of an oscillation in traditional EOF analysis (Monahan and Dai, 2004; Maher et al., 2018). For comparison, we also analyze the time evolution of the ensemble-based ENSO Niño3 amplitude defined by the ensemble standard deviation of Niño3 index in the [5°S, 5°N]×[150°W, 90°W] Niño3 region.

Note that PC1 is actually an index for the ENSO phase. It is found to be closely related to the standard Niño3 index with a correlation coefficient of 0.98 (Ashok et al., 2007), which confirms that the first mode of EOF represents the conventional ENSO well. It has been used in this quality in other studies as well, see, e.g., Diaz et al. (2001) who carried out traditional EOF analysis using a slightly smaller Pacific region. As an index, its main use is a standardized indication for the state of the remote phenomena related to ENSO. Even though nowadays more complex indices exist for the characterization of ENSO, created, e.g., by the combination of PC1 and PC2 (such as the ones in Takahashi et al. (2011)), we choose PC1 to provide a simple and easy-to-follow example on illustrating the applicability and advantages of the SEOF analysis.

For characterizing such teleconnections, an instantaneous ensemble-based correlation coefficient can be determined between the PC1 and PRECT data of the ensemble members at each time instant and for each grid point. As ENSO has its maximum around boreal winter, which is traditionally defined as DJF, we analyze the DJF ENSO pattern and ENSO teleconnections in the paper. In order to investigate the possibility of predicting precipitation half a year in advance based on PC1, we calculate lagged correlations beyond instantaneous ones. The relationship between ENSO and the South Asian monsoon is believed to be one of the most important teleconnection phenomena and is traditionally investigated using JJAS (Krishna Kumar et al., 1999; Ashok et al., 2007; Srivastava et al., 2019, see, e.g.), and West Africa also receives the major proportion of its annual rainfall in JJAS (Srivastava et al., 2019), therefore, we utilize both DJF-mean and JJAS-mean data in order to reveal simultaneous and lagged connections, and correlate the JJAS PC1 with JJAS PRECT, the DJF PC1 with DJF PRECT, the DJF PC1 with JJAS PRECT, and JJAS PC1 with DJF PRECT over the ensemble. The choice of DJF and JJAS seasons is also used in Wu et al. (2012) for studying the ENSO influences on Indian summer monsoon. The investigation of lagged correlations opens the possibility of studying the predictability of the amount of precipitation in different regions based on the PC1.

## 2.3  Comparing the capabilities of the snapshot and traditional methods

To better understand the facilities of the snapshot methods in general, it is worth giving an overview of the advantages and limitations of both the snapshot methods and traditional temporal ones. When only single time series are available, such as measurements or single time series of model simulations, one has to use the traditional temporal methods of time series analysis to investigate a phenomenon. For example, in the case of ENSO when examining its characteristics by EOF analysis, used in

this study as well, it means that for the time period of the analysis a single EOF loading pattern is obtained, i.e., the spatial pattern of the oscillation must be assumed to be constant for the given time period. The phases of this "standing" oscillation over time are then provided by the corresponding PC1s, and the strength of ENSO teleconnections in the studied time period can be characterized at every geographical location by a single correlation coefficient between the PC1 time series and the local time series of the studied quantity. However, during this time period (e.g., within 30 or 100 years) climate changes (excluding the non-realistic case of stationary forcing). For example, in the case of the CESM-LE climate change indeed manifests in a considerable global surface temperature increase on decadal time scale between 1950 and 2100 (Kay et al., 2015). Therefore, neither it can be presupposed that the pattern of the ENSO or the strength of its teleconnections remain constant during climate change, nor it can be assumed that a single oscillation pattern or correlation coefficient can faithfully characterize the conditions of several years.

A changing climate may manifest even in local regime shifts and appearance of tipping cascades in such coupled system (Klose et al., 2019). For a demonstration of the possibility of such cascading events in the climate system in a conceptual model of a coupled North Atlantic Ocean – ENSO system, see Dekker et al. (2018). These transitions may cause qualitative changes also in local dynamics and in the connections of the climate system. Climate change can also result in chaotic synchronization between certain regions which behave as chaotic oscillators (Duane, 1997), e.g., implying a change in the relationship between the Atlantic and Pacific sectors (Duane and Tribbia, 2001). As another example, Falasca et al. (2019) showed both in the CESM-LE and in reanalysis data that the connection between the Equatorial Atlantic region and the ENSO may exist only in certain decades, possibly because of chaotic synchronization.

Naturally, the analyzed time period can be divided into smaller time windows, and then some kind of time-dependence of the oscillation and teleconnections can be studied. However, if the forcing is non-stationary, the climate can change even within shorter time periods. The smaller the time window the less the climate is expected to change, but meanwhile, the less robust the calculated statistics will be due to the smaller number of temporal data points. Regarding the case of changing climate, the optimal choice would obviously be to choose a series of time windows of length of one data point within which the climate evidently cannot change. This is exactly what the snapshot methods do by calculating statistics at single time instants; however, an ensemble is needed for this purpose. Therefore, the methods of the snapshot framework operate across the ensemble dimension of climate realizations at single time instants, and provide measures and statistics that characterize only the instantaneous potential states of the climate system without a direct impact of previous or future climate states on the value of the statistics, in contrast to results from traditional time series analysis operating along the time dimension. For example, the most known ensemble statistics, the ensemble mean and ensemble standard deviation describe the mean state and the strength of the internal variability of the possibilities permitted by the instantaneous climate, respectively, at the chosen time instant under the external forcing history up to that time. Analogously, an instantaneous ensemble-based SEOF loading pattern represents the spatial structure of an oscillation that characterizes the potential variability of the climate states of the given time instant, and the corresponding PC1s reveal the phases in which the ensemble members are in that very moment.

A similar overview of comparing the results derived from snapshot methods and time series analysis can be found in Herein et al. (2017) and Bódai et al. (2019) on the example of the North Atlantic Oscillation teleconnections using a station-based NAO

index and the ENSO phenomenon using the Niño3 and SOI indices. The single time series results were shown to be strongly different from the snapshot ones. These papers also illustrate by numerical examples that the choice of the time window may have a considerable effect on the statistical measures in the traditional approach, while this is not a problem using the snapshot framework.

As an obvious limitation, to take the advantage of the methods within the snapshot framework on providing a mathematically correct way to describe the instantaneous statistics and characteristics of the potential climate states under the external forcing history up to that time, an ensemble of climate realizations is needed. The more reliable and robust statistics are aimed to be obtained, the larger ensembles are required. Note that the number of temporal data points is also an issue in the traditional time series analysis as well. We also feel important to repeat here that the desirable tools of traditional time series analysis give robust and unbiased results only if the underlying statistics can be approximated well as stationary. Furthermore, temporal autocorrelation may seriously reduce the effective sample size of the traditional methodology: already with a 3-year autocorrelation, the effective length of a 150-year time series is practically not more than 50 data points.

Most often, an ensemble is produced by a climate model, however, they may also be accessible in experiments. For example, they proved to be useful in laboratory experiments aiming to study the effect of climate change on mid-latitude atmospheric circulation (Vincze et al., 2017). Obviously, the obtained results are constrained by the climate model or the capability of the experimental setup, while the tools of time series analysis can be applied also to historical measurements or reanalysis data, i.e., for cases when only single time series exist due to the fact of having only one Earth history.

## 3 Results

### 3.1 Changes in the ENSO pattern and amplitude

For a first impression of the SEOF analysis, the instantaneous ensemble-based regression maps of the first SEOF mode for the SST in the Pacific region for the JJAS period and DJF period are shown in Fig 1.a-c and Fig 1.d-f, respectively. As expected from observation-based data (see, e.g., Kirtman and Shukla, 2000), the typical amplitudes of the SST anomaly values across the ensemble members at the Equatorial Pacific are somewhat larger in DJF than in JJAS. The shape of the pattern clearly changes somewhat over time and the explained variance of the first SEOF mode also varies.

To determine whether the observed alterations are due to fluctuations because of the finite number of ensemble members or the consequences of the changing climate, a linear fit over time is performed on the ensemble-based regression maps at each grid point from 1950 to 2100 (Fig 2). The applicability of linear fit to the time series of regression coefficients is checked at two random grid points in the region of strong positive trends and two in the region of negative trends in Fig. 2 for both seasons (see Fig. S1 in the Supplement), and the trend of the curves is found to be well approximated by linear regression. In the regression maps for JJAS (Fig 2.a) a positive trend with $[1\text{--}3]\times10^{-3}\ ^\circ\text{C}\,\text{yr}^{-1}$ can be detected in the Niño3 and Niño3.4 region, while close to Indonesia and Australia a clear negative trend with the same magnitude appears. It corresponds to an increase of 0.15 to 0.45 $^\circ$C in the SST variability, respectively, in 150 years (from 1950 to 2100), which is a considerable change compared to the magnitude of 0.1–1 $^\circ$C in the SST variability in Fig. 1. For DJF (Fig 2.b) a somewhat narrower band with slightly weaker increase of $[0.5\text{--}2]\times10^{-3}\ ^\circ\text{C}\,\text{yr}^{-1}$ can be seen all along the Equator in the Pacific Ocean and negative trends in similar

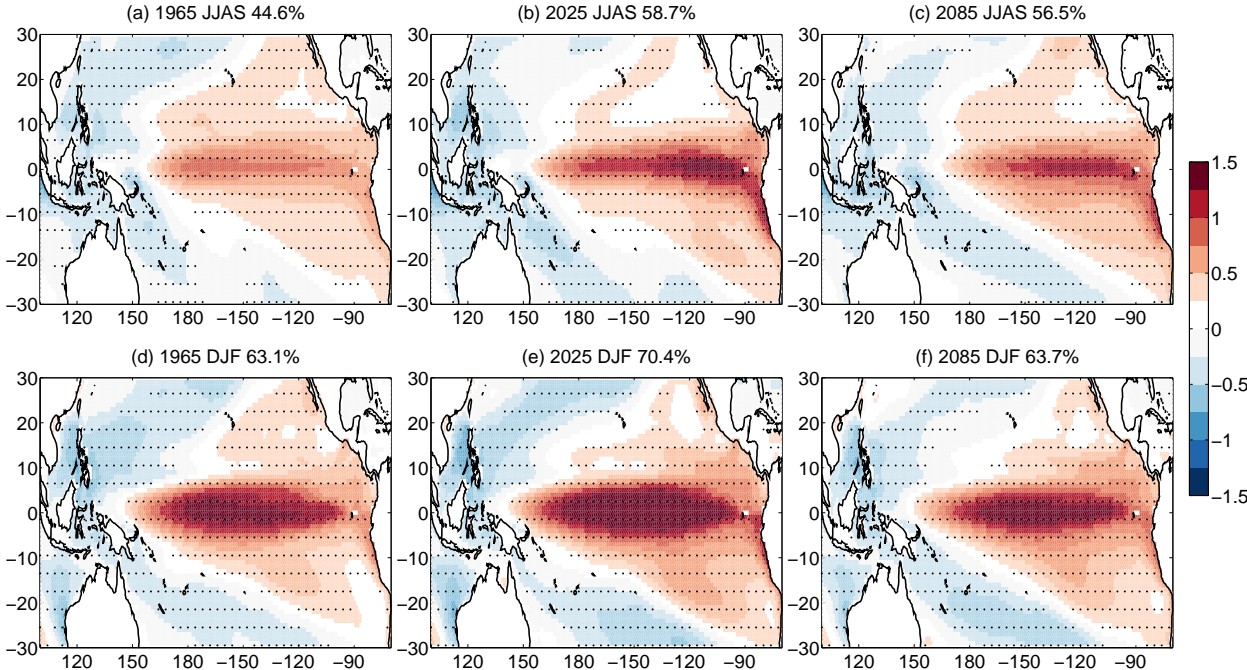

**Figure 1.** Ensemble-based SST regression maps [°C] for years given in the panel's title for JJAS (a-c) and DJF (d-f). The explained variance of the first SEOF mode is also displayed in the title of the panels. Dots represent geographical locations where the regression coefficient is significant at the 95% level. For better visibility, only every fourth grid point is dotted.

magnitude appear near Australia and at the western coast of South America. Similar trends in the annual ENSO pattern were pointed out in Maher et al. (2018) by EOF-E analysis mentioned in the Introduction, however, Fig. 2 draws attention to the fact that these patterns also have seasonal dependence. Ensemble-based instantaneous regression maps present typical value of the

amplitude of the fluctuations directly related to the given EOF mode of variability at each grid point, which in the case of EOF1 has the strongest relationship with ENSO. The temporal changes in the regression maps are then easy to interpret intuitively: they show the changes in the fluctuation amplitudes, i.e., changes in the typical SST anomalies bound to the given mode at each grid point, and potential shifting in the pattern during climate change as well. Nevertheless, it is also worth studying an other representation of the ENSO pattern, the pattern which is separated from the amplitude of the phenomenon. The time

series of this separated pattern, represented by the loading pattern of the SEOF analysis, and the changes in them are shown in Fig. S3 and Fig. S4 in the Supplement, respectively. The temporal changes in these normalized patterns show the alterations in the relative importance of different regions from the point of view of the given mode. Their patterns in Fig. S4 are similar to the ones in Fig. 2, however, in contrast to them, no dominant maximum can be seen in the Niño3 region for DJF. Furthermore, while the amplitude increase in the Niño3 region in Fig. 2.a in JJAS proves to be more pronounced than the decrease in the

western part of the equatorial Pacific, Fig. S4 shows that the reduction of the relative importance of the latter is greater than the former one.

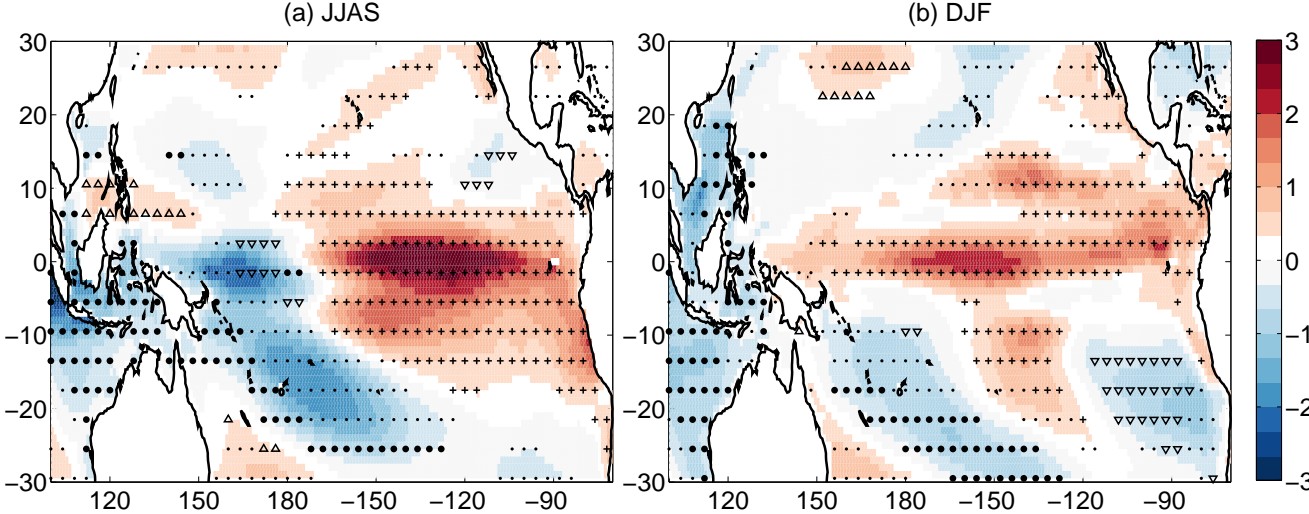

**Figure 2.** The slope of the linear fit [$10^{-3}$ °C yr$^{-1}$] at each grid point of the (a) JJAS and (b) DJF regression maps from 1950 to 2100. Dots represent geographical locations where the trend is significant at the 95% level. Where, additionally, (i) the trend is positive and the regression coefficients are positive and significant at the 95% level in the temporal mean "+" signs are displayed, (ii) the trend is positive and the regression coefficients are negative and significant at the 95% level in the temporal mean "△" signs are displayed, (iii) the trend is negative and the regression coefficients are positive and significant at the 95% level in the temporal mean "▽" signs are displayed, and (iv) the trend is negative and the regression coefficients are negative and significant at the 95% level in the temporal mean "●" signs are displayed, respectively. For better visibility, the significance and direction of the trend is indicated at only every fourth grid point.

Another particularly important feature of the ENSO phenomenon is the ENSO amplitude, which shows a large diversity in different climate projections (Yeh and Kirtman, 2007; Collins et al., 2010; Chen et al., 2015). Therefore, besides the exploration of the changes in the ENSO pattern, we also quantify the potential changes in the ensemble-based instantaneous ENSO strength
(the ensemble standard deviation of the PC1 ($\sigma_{\mathrm{PC}}$)) and the change in the explained variance of the first SEOF mode (Fig. 3.a,d, and b,e). Furthermore, the ensemble-based analog of a traditional amplitude of the ENSO phenomenon, namely, the Niño3 amplitude is also studied (Fig. 3.c, f). The value of $\sigma_{\mathrm{PC}}$ is much smaller in JJAS (Fig. 3.a) than in DJF (Fig. 3.d), and Fig 3.b and d also show that the explained variance in the SST variability by the first SEOF mode is about 15% greater in DJF than is JJAS. A systematic increase is found in all three quantities both for JJAS and for DJF. The increase in JJAS is around 20% in
the $\sigma_{\mathrm{PC}}$ (Fig. 3.a), in the explained variance (Fig. 3.b), as well as in the ENSO amplitude (Fig. 3.c), while the increase in DJF is somewhat lower, approximately 5-15% for the three quantities. The larger values of the explained variance mean that by 2100 the oscillation associated with the first mode is going to be responsible for a much larger fraction of the variability in the SST fields. The increasing values in the explained variance of the first SEOF mode are found to be compensated by the generally slightly decreasing trends appearing in the explained variance of higher-order modes (see Fig. S2 in the Supplement). In
JJAS, the second mode contributes the most to the decrease by 2.4%, while in DJF, for which Fig. 3.e shows a less pronounced

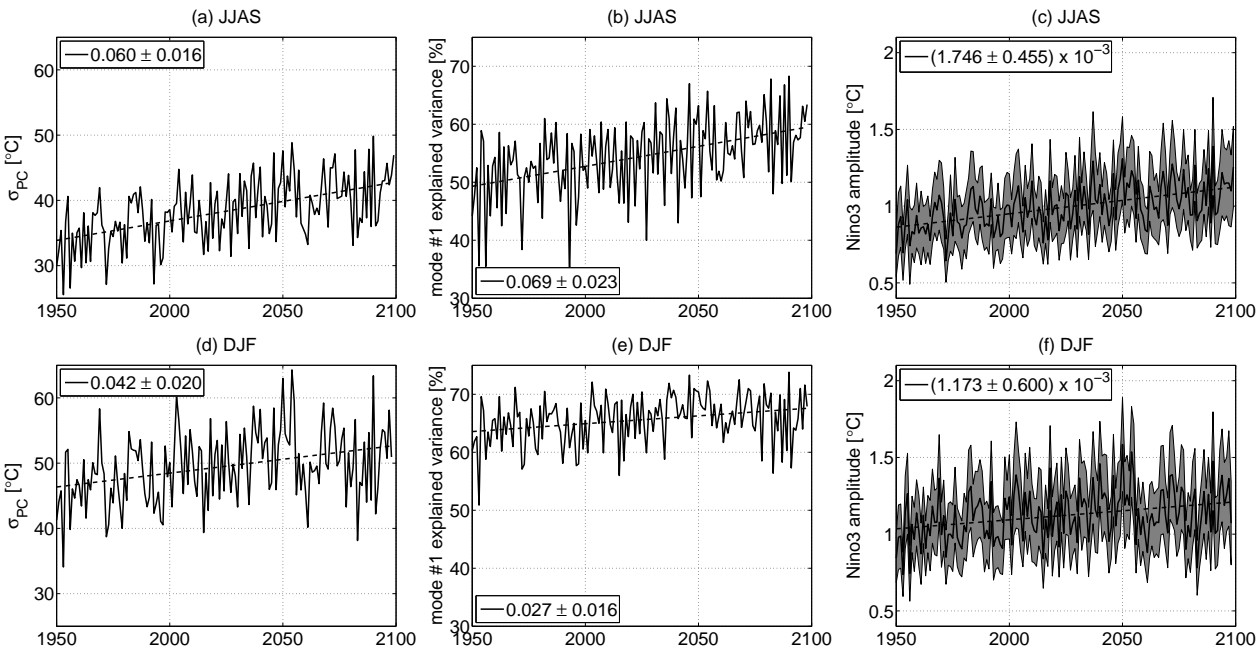

**Figure 3.** (a) and (d) ensemble-based ENSO strength as the ensemble standard deviation of the PC1 ($\sigma_{\mathrm{PC}}$), (b) and (e) the explained variance of the first SEOF mode, and (c) and (f) Niño3 amplitude as the area-mean (thick line) ensemble standard deviation of SST in the Niño3 region and its area standard deviation (grey band)) in (a)-(c) JJAS and in (d)-(f) DJF. Linear fits are indicated by dashed lines, legends indicate the slope of the linear fits with 95% confidence intervals.

increase for the first mode, the explained variance of the second mode is approximately constant, and the compensating decrease appears in the explained variance of the higher-order modes. For more details, see Fig. S2 in the Supplement. The approximately 20% increase in the Niño3 amplitude is in fairly good agreement with the study of Maher et al. (2018). It is comparable with the approximately 10% increase within 100 years in Zheng et al. (2018) found for the CESM-LE for the RCP8.5 scenario using
sliding windows temporal statistics. This result reveals that the forced response of the ENSO amplitude under climate change is positive. Obviously this finding is valid for the CESM-LE only, while other models may behave differently regarding the ENSO amplitude (Yeh and Kirtman, 2007; Kim et al., 2014; Chen et al., 2015). We mention that the first ensemble-based study (which investigated several large ensembles) reported that an increase or zero trend is likely regarding the Niño3 amplitude change (Maher et al., 2018).

**3.2   Changes in ENSO's teleconnections**

To study the potential changes in the ENSO-related precipitation events over time, instantaneous ensemble-based correlation coefficients ($r$) between the PC1 and the total precipitation PRECT at each grid point are determined for both zero-lag and plus half-year-lagged PRECT data.

In Fig. 4 these $r$ maps are presented. Four different combinations of PC1 and PRECT correlations are analyzed. The simultaneous (zero lag) correlation between the JJAS data in Fig 4.a-c shows that there are places where the correlation is remarkably negative, indicating dryer conditions during warm events and wetter weather during cold episodes. It is typically observable for Indonesia and Australia ($r \approx (-0.5) - (-0.8)$), East Africa ($r \approx (-0.4) - (-0.6)$), Central America ($r \approx (-0.4) - (-0.6)$), the northern part of South America ($r \approx -0.4.. - 0.6$), and southern and western part of India ($r \approx -0.4$). At the same time positive correlations, indicating more than average precipitation during El Niño and less than average for La Niña, can be seen for North Africa ($r \approx 0.4$) and the West Coast of the USA ($r \approx 0.3 - 0.6$). The above-presented picture is roughly consistent with observation-based investigations, see, e.g., Diaz et al. (2001).

It is worth assessing other variations of PC1 and PRECT correlations. The zero-lag correlation between DJF PC1 and DJF PRECT can be seen in Fig. 4.d-f. The anti-correlation with $r \approx (-0.4) - (-0.8)$ for South India, Indonesia and Australia still holds, as well as the positive correlation for the West Coast of the USA, while the East African correlation changes sign. In general, the DJF correlation patterns are also fairly close to the observation based ones, which was reported, e.g., in Diaz et al. (2001); Yang and DelSole (2012); Yeh et al. (2018).

The role of the lagged correlations is also a relevant issue. It is known that lagged correlation is especially essential, e.g., for the Indian summer monsoon due to the potential monsoon forecasting (Wu et al., 2012; Johnson et al., 2017; Kucharski and Abid, 2017). Therefore, we construct the ensemble-based lagged correlation maps using the DJF PC1 and JJAS PRECT correlation (Fig. 4.g-i). The negative correlation appearing in observations for Indian JJAS precipitation with DJF ENSO phase (Wu et al., 2012) seems to be quite weak in CESM-LE, only a part of South India has some anti-correlation. This means that in CESM-LE the Indian summer monsoon does not seem to be predictable by the DJF PC1 of the ENSO. In contrast to this, as in the previous cases, the amount of the JJAS Indonesian and Australian precipitation shows a clear negative relationship with the DJF PC1. Examining the correlation between the JJAS PC1 and the following DJF PRECT we get interesting results (Fig. 4.j-l). India, the Indochinese Peninsula and Indonesia show a quite strong anti-correlation ($r \approx (-0.4) - (-0.8)$), while from the northern part of the Indian Ocean through the eastern and central part of Africa to northward to the Arabian Peninsula a considerable positive correlation can be observed. This suggests that based on the JJAS PC1, as an index for the actual ENSO phase, the DJF precipitation conditions are mostly predictable in these regions.

The question naturally arises, whether climate change has an impact on the strength of the teleconnections in these regions. To answer the question, similarly to the trends obtained for the regression maps in Fig. 2, we construct global linear trend maps of the ensemble-based correlation coefficient maps over time (Fig. 5). For the zero-lag correlations in JJAS (Fig. 5.a), among those regions where the strength of the connections are high during the investigated 150 years (indicated by different markers), Australia, an extended part of Indonesia and the southern part of South America show a clear increase of $[1\text{-}2] \times 10^{-3}$ yr$^{-1}$ in the anti-correlation, while some spots in the southern part of the Atlantic Ocean and Central Africa have an increasing positive correlation during 1950–2100, that is, a strengthening of the connection between the phase of the ENSO and the precipitation conditions is found for these regions. South India's and Central America's positive correlation, the Phillipine Sea's negative correlation seem to weaken. For the zero-lag correlation in DJF (Fig. 5.b) a somewhat different picture opens up. The strength of the positive correlation from Central Africa to the Aral Sea and in the North Atlantic Ocean and the strength of the negative

correlation in South India increase further by a trend of $[1\text{-}4]\times10^{-3}$ yr$^{-1}$, and in the Niño3 region a somewhat stronger positive
trend can be found than in JJAS. The spatial distribution of the correlation coefficients is similar for the JJAS PC1 and DJF
PRECT (Fig. 2.d). The lagged correlations for DJF PC1 and JJAS PRECT (Fig. 5.c) are found to increase considerably near the
eastern coast of Africa, in the Niño3 and Niño4 regions and around the Caribbean Islands. Thus, based on the larger value of
the correlation coefficient implying a stronger relationship, we conclude that a half-year-forward estimate of the precipitation
from PC1 data in these regions becomes more accurate. The above-mentioned $[1\text{-}4]\times10^{-3}$ yr$^{-1}$ trends lead to a remarkable
change of 0.15-0.45 in the $r$ values during 150 years.

## 4    Conclusions

In this study, we investigated the changes in the ENSO phenomenon and the alterations of its precipitation-related teleconnec-
tions for 1950–2100 in the CESM-LE climate simulations. To avoid the disadvantages of the subjective choices of traditional
temporal methods, here we used the ensemble-based snapshot framework providing instantaneous quantities computed over
the ensemble dimension of the simulations. To our knowledge, this is the first time when the snapshot empirical orthogonal
function (SEOF) analysis using SST data is utilized to reveal changes in the pattern and amplitude of the ENSO. Instantaneous
ensemble-based correlation coefficients between the principal components (PC1s) of the first SEOF mode (considered here as
an index for the ENSO phase) and the total precipitation at each grid point over the Globe were also determined to evaluate the
ENSO's precipitation-related teleconnections detailed below.

Our results show that the ENSO pattern undergoes remarkable changes during the investigated time period. This is found to
be more pronounced in the JJAS season, where the ensemble-based SST regression maps show even 0.45 °C and $-0.45$ °C
change in the Niño3–Niño3.4 region and in the western part of the Pacific Ocean over 150 years, respectively. We note that
these changes are of the same order of magnitude as the typical SST variability across the ensemble at different time instants,
which is found to be of 0.5–1.5 °C in the equatorial region. The Niño3 amplitude also increases by about 20% and 10% in JJAS
and DJF, respectively. We found a clear growth of similar rate also in the ENSO strength (defined as the ensemble standard
deviation of the PCs) and in the explained variance of the first SEOF mode. This means that the amplitude of the fluctuations in
the SST field will increase, and the first SEOF mode will explain a much larger fraction of the variability of the SST fields by
the end of the 21th century. In general, a larger change in the different quantities is found for JJAS than for DJF. While at the
beginning of the JJAS season the ENSO cycle is generally just switching phase in the CESM-LE (Wieners et al., 2019), DJF
can be considered to be the "main" ENSO season with the largest SST anomalies. The smaller changes in the DJF quantities
may be explained by the conjecture that, calculated for the main ENSO season, the DJF characteristics may be more robust
and, thus, undergo weaker alterations during the investigated 150 years than the JJAS ones, which are calculated around the
phase change of the cycle. A more thorough investigation of this question could be a topic of future research.

The precipitation-related teleconnections of the ENSO also show a considerable change over time in several regions. For
example, the anti-correlation with precipitation in Australia and in the southern edge of South America in JJAS are predicted
to be more pronounced by the end of the 21th century, as it changes from about $-0.5$ by $-0.15$. At the same time, the positive

correlation in Central Africa and the western coast of South America, especially in Chile, becomes enhanced by $0.15 - 0.3$ as well.

Lagged correlation coefficients reveal potential predictability of the precipitation conditions based on ENSO's PC1. We found that the amount of precipitation in Australia, New Zealand, Indonesia in JJAS is generally less than average after DJF warm episodes, while, e.g., the eastern coast of Africa is wetter than average. In DJF the amount of precipitation in Australia and in South India is less than average after previous warm conditions in JJAS, however, the central islands in Indonesia and a large part of East and Central Africa get more precipitation. Our results show that the strength of these connections strengthens over time, especially in the African region up to the Arabian Peninsula, and slightly in South India.

As an outlook, we mention that according to Seager et al. (2019) in most of the state-of-the-art climate models the west-to-east warm-to-cool SST gradient decreases with rising greenhouse gas concentration, inconsistent with reanalysis data. They found that similarly to most of CMIP5 models, the ensemble mean of CESM-LE also shows a moderate SST trend over 60 years in the Niño3.4 region inconsistent with HadISST (Rayner et al., 2003) and NCEP/NCAR reanalysis (Kistler et al., 2001). However, this trend proves to be smaller than the CMIP5 multimodel mean for the studied time interval for end years 2008—2017, and some of the ensemble members approach well the values derived from reanalysis. Furthermore, the ECMWF/ORAS4 reanalysis (Balmaseda et al., 2013) trend values are quite close to the CESM-LE ensemble mean for end years of 2008—2009. The deviation in SST trend between the observations and CESM-LE may have an effect on the strength of and change in the teleconnections, however, since Section 3.2 proves that the results from CESM-LE obtained by SEOF analysis are roughly consistent with the observed teleconnections and the CESM-LE performs relatively well according to Seager et al. (2019) compared to the most of CMIP5 models, we expect that it does not influence much the strength and changes of the connections found in this study.

Finally, we note that our snapshot method lacks any temporal statistics for the EOF analysis and correlation calculation for revealing teleconnections; thus, it is an objective way to explore the time evolution of other phenomena and teleconnections during climate change. A larger number of ensemble members may even result in more accurate statistics and smaller fluctuations in the time series.

*Code availability.* Codes are available from T.H. on reasonable request.

*Author contributions.* T.H., M.H., T.B. conceived of the presented idea. M.H. downloaded and preprocessed the data, T.H. worked out the technical details, performed the computations and plotted the results. T.H. and M.H. took the lead in writing the manuscript. All authors discussed the results and contributed to the final manuscript.

*Competing interests.* The authors declare no competing interest.

*Acknowledgements.* Fruitful discussions with G. Drótos, T. Tél and M. Vincze are gratefully acknowledged. This paper was supported by the János Bolyai Research Scholarship of the Hungarian Academy of Sciences (T. H.), by the National Research, Development and Innovation Office – NKFIH under grants PD-121305 and PD-132709 (T. H.), PD-124272 (M. H.), FK-124256 and K-125171 (T. H., M. H.) and by the Institute for Basic Science (IBS), South Korea, under grant IBS-R028-D1 (T. B.). The authors also wish to thank the Climate Data Gateway at NCAR for providing access to the output of the CESM-LE. The CESM-LE output is available at http://www.cesm.ucar.edu/projects/community-projects/LENS/data-sets.html.

370

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

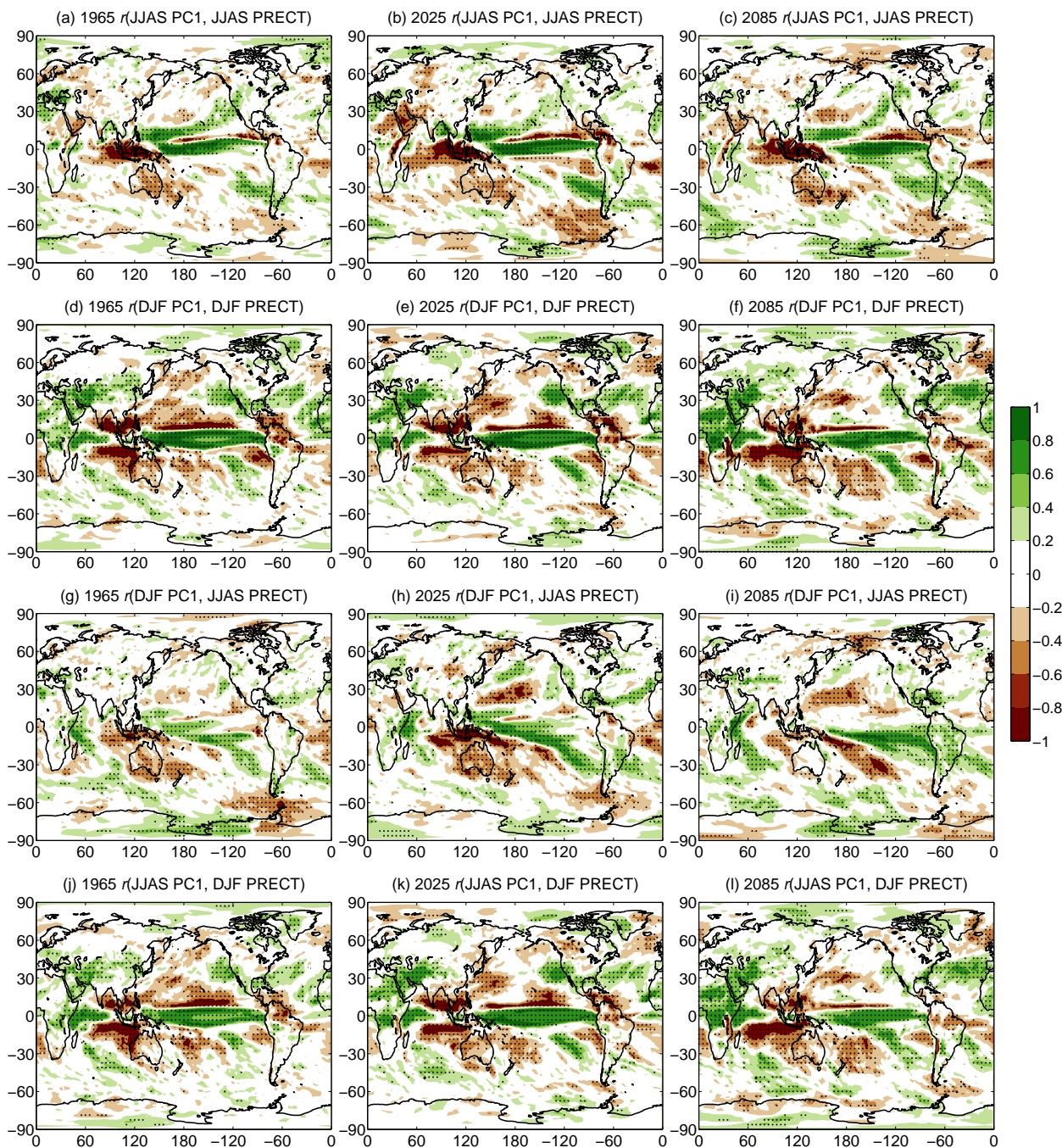

**Figure 4.** Ensemble-based correlation coefficient *r* maps for the JJAS PC1 and JJAS PRECT (a-c), DJF PC1 and DJF PRECT (d-f), DJF PC1 and JJAS PRECT (g-i), JJAS PC1 and DJF PRECT (j-l). Specific years are indicated in the panels. Dots represent geographical locations where the correlation coefficient is significant at the 95% level. For better visibility, only every fourth grid point is dotted.

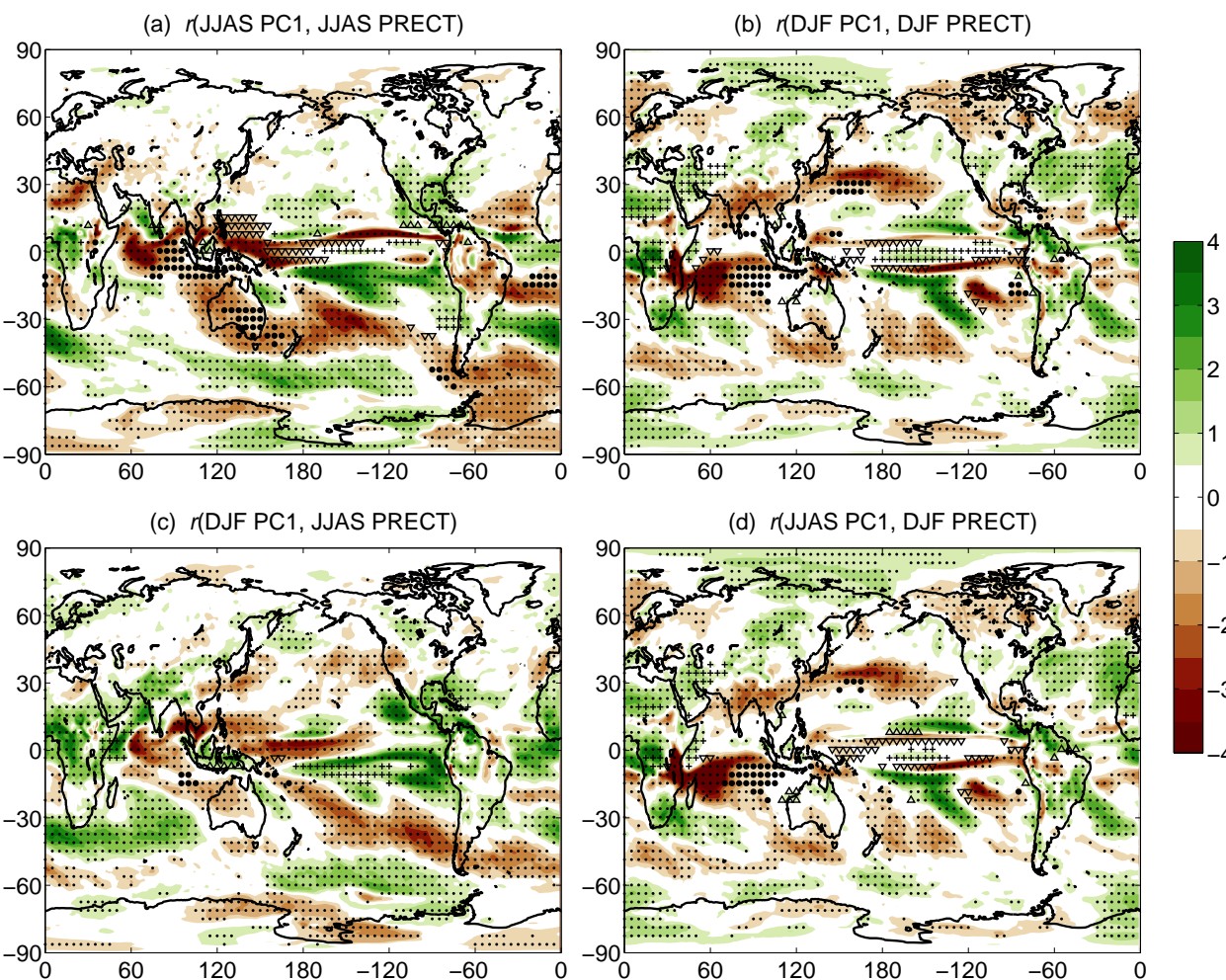

**Figure 5.** The slope of the linear fits $[10^{-3} \text{ yr}^{-1}]$ at each grid point for the correlation coefficient $r$ for the JJAS PC1 and JJAS PRECT (a), DJF PC1 and DJF PRECT (b), DJF PC1 and JJAS PRECT (c), JJAS PC1 and DJF PRECT (d). Dots represent geographical locations where the trend is significant at the 95% level. Where, additionally, (i) the trend is positive and the correlation coefficients are positive and significant at the 95% level in the temporal mean "+" signs are displayed, (ii) the trend is positive and the correlation coefficients are negative and significant at the 95% level in the temporal mean "△" signs are displayed, (iii) the trend is negative and the correlation coefficients are positive and significant at the 95% level in the temporal mean "▽" signs are displayed, and (iv) the trend is negative and the correlation coefficients are negative and significant at the 95% level in the temporal mean "●" signs are displayed, respectively. For better visibility, the significance and direction of the trend is indicated at only every fourth grid point.