# Peer review of "On the time evolution of ENSO and its teleconnections in an ensemble view – a new perspective"

_Earth System Dynamics, 2019_

## Referee Comment (RC1) · Anonymous Referee #1 · 4 Nov 2019

The authors investigate changes in the ENSO phenomenon and its teleconnections using a recently proposed methodology (SEOF). ENSO is the most dominant interannual mode of variability in the climate system: studying changes in its amplitude and its impacts is then relevant to the scientific community. Differently from standard statistical methods in time series analysis, the authors used a new methodology, snapshot empirical orthogonal function (SEOF). SEOF is tailored for ensemble simulations: after a transient time, each member of an ensemble of simulations of a nonlinear dynamical system is believed to cover the distribution of states of the attractor. The authors exploit this to define a new notion of Empirical Orthogonal Function (EOF) along ensemble members, rather than along the time dimension. This methodology allows for the definition of "instantaneous" ENSO patterns. Here the authors consider as "instantaneous"

[Figure]

seasonal averages (JJAS and DJF) and proceed in analyzing (a) ENSO pattern and its evolution in time and (b) changes in teleconnections, using the proposed methodology. Changes in the spatial pattern of ENSO (quantified by a regression of the PC of the first SEOF mode on SST) are briefly examined qualitatively and quantitatively by linear fitting the regression maps. Interestingly, the largest changes in patterns are in the JJAS season rather than in DJF. Changes in amplitude of ENSO are also examined. In agreement with previous studies, the authors find an increase in variance of the PC1 and of the Niño 3.4 index. Finally, the authors use the proposed framework to investigate changes in the teleconnection patterns between ENSO and precipitations, using instantaneous and lagged correlations along ensemble members, rather than along the time dimension, and explore the changes of these correlations with time.

This is a well written, clear and interesting paper. Also, the application of the SEOF methodology on ENSO is new. I have some minor comments, mainly regarding the methodology.

(a) (a) The main strength of this methodology is that it allows to analyze large ensembles in a comprehensive and well defined way. Also this framework offers a route to examine teleconnections, disentangling climate variability and external forcing in a correct way. However, in my opinion, this does not mean that this method is definitively better than traditional time series analysis. Both this methodology and temporal statistics are useful for different reasons. Here some reasons:

- It is true that the choice of the time window is largely subjective. However, ENSO has a quasi-periodicity of 3 to 7 years and its teleconnections can be analyzed a 12-months range (e.g., ENSO leads the Western Indian Ocean with a lead lag of ∼3 months). I expect that, given a single member, time windows from 30 to 100 years of data would give robust results. If correlations between two basins, start changing when considering 30 or 100 years it can simply mean that the connections analyzed may not be "stable". This is possible in climate and can be a result of (i) local regime shifts in one of the two basins, causing qualitative

changes in local dynamics and so in connections with other basins (see Dekker et al. https://www.earth-syst-dynam.net/9/1243/2018/esd-9-1243-2018.pdf or Klose et al. https://arxiv.org/pdf/1910.12042.pdf) and/or (ii) phenomena of chaotic synchronization between basins (please see this PRL paper from Duane and Tribbia https://pdfs.semanticscholar.org/cd01/9dacfa47fdc2d5b46e8d33dda956fae135b0.pdf). An example of a (possibly) unstable teleconnections is the leading from the Equatorial Atlantic to ENSO. For example, Falasca et al. (https://agupubs.onlinelibrary.wiley.com/doi/epdf/10.1029/2019MS001654) showed that this lead may exist only in certain decades (see Figures 18e, 18f for reanalyses and Figures 19e and 19f for two CESM members), possibly for phenomena of chaotic synchronization. In the case of this specific teleconnection choosing a window of 50 or 100 years would indeed give a different result, but not because of biases in the methodology but because this connection seems to change in time. Also, in the context of the CESM-LE it has been found at different times in different members, suggesting indeed a chaotic synchronization between the two basins.

- Traditional time series analysis (referred in the paper as "temporal statistics") presents lots of desirable tools: (i) different measures of coupling between time series such as linear (e.g., Pearson correlation), nonlinear (e.g., Mutual Information), causal (e.g., PCMCI algorithm) and (ii) robust methodologies to assess statistical significance. This is possible if a large number of data points is analyzed. In the snapshot method, every measure of coupling and every test is constrained by the (very small) number of members of an ensemble. This is a limitation of the methodology since 40 members is still a (very) limited number of data points in the analysis.

- More importantly, all results of the snapshot methodology, live in model-land (see http://www.economics-ejournal.org/economics/discussionpapers/2019-23/file). In fact, in reality we have only access to one climate and we have no access to an ensemble. Therefore, the results that can be obtained using this methodology, while interesting, are always going to be constrained to the chosen climate model and its biases.

These points should be briefly discuss. Advantages and disadvantages of both the snapshot methodology and traditional time series analysis should be made clear. My view is that they are both useful and can complement each other, and not that one is definitive better than the other.

(b) Figure 2. Are the trends of the regression maps really linear? Was this checked? I would have expected to be linear in a time range of ∼30 years but not necessarily from 1950 and 2100. Can you please check two random time series in the ENSO region and in the Horse Shoe Pattern (the region with strong negative linear trend) in Fig. 2a and see the shape of the trend?

(c) Figure 3. Panels (b) and (e). It is interesting to see that while the explained variance of PC1 in DJF is relatively constant, this is not true for the season JJAS. In Figure 3b the explained variance of the PC1 experiences a steady increase from ∼45% to ∼60%. It could be interesting to analyze the second mode of the SEOF and see how it is changing. If this analysis would help in better understanding (or at least suggests an explanation) the increase in variance of the first mode I would recommend to add the analysis of the second mode in the appendix.

(d) Figure 5. Second line of the caption. Correct: DJF PRECT (b → DJF PRECT (b)

---

## Short Comment (SC1) · 12 Nov 2019

Remarks on 'On the time evolution of ENSO and its teleconnections in an ensemble view – a new perspective'

This manuscript provides some interesting ideas by building on previous work that used the ensemble dimension in a large ensemble to describe forced changes in the statistics of the climate system. In particular changes in teleconnections, here characterised by the correlation coefficient in the ensemble dimension, may provide some new insights.

I was wondering why you decided to focus on the first EOF to characterise ENSO variability. Takahashi et al. (2011) argue that both EOF1 and EOF2 should be used

to characterise ENSO. Did you test if there are changes in the second EOF? Changes in EOF 2 might also project on the Nino3 region and could theoretically even have an opposing effect compared to the changes in EOF 1 discussed in this manuscript.

How much does the sampling uncertainty affect the detected changes? I.e. how much of the difference in variance between two years can be attributed to the forcing change and how much of the difference is due to the limited ensemble size? Note that we concluded in Maher et al. (2018) that 30-40 ensemble members are sufficient to quantify ENSO variability when analysing ENSO variability over time periods of 10-50 years. Arguably, 10 years and 30 members might not even be sufficient, depending on the acceptable error (figure 4 in Maher et al. 2018). Since you are using indvidual years, it could be possible that more than 30-40 members are required. Based on this, I would expect to see large sampling uncertainty in the correlation coefficients. It might be beneficial to show the time series for the correlation coefficients for some selected regions to demonstrate that the discussed changes are larger than the sampling uncertainty.

Separating amplitude and pattern changes: In figure 1, you standardised the PC1. Thus both pattern and amplitude changes, if they occur, can be seen in the regression maps. Did you use the same approach for the analysis in figure 3? An alternative approach to separate pattern and amplitude changes would be to normalise the pattern. Amplitude changes can then be seen in the PC, whereas pattern changes can be seen by comparing the regression maps for different states of the climate system. This is the approach we used in Maher et al. (2018).

---

## Referee Comment (RC2) · Anonymous Referee #2 · 13 Nov 2019

**Review comments: On the time evolution of ENSO and its teleconnections in an ensemble view - a new perspective (ESD-2019-57)**

In this paper, the authors aimed to examine changes in ENSO SSTA patterns and teleconnections under climate change, using a recently developed ensemble-based method (SEOF). SEOF was applied on all the ensemble members from CESM1 at each time step, avoiding using statistical mean or standard deviation under a non-stationary climate.

This ensemble-based method provides a very interesting perspective to study change in the warming climate. This manuscript showed how to apply this method on the change in ENSO patterns. However, I do feel this manuscript lacks details in terms of the physical interpretation of the method (SEOF) and results, which makes the article quite difficult to follow. Also, the content of change in teleconnections which only used correlation to analyze (Section3.2) seemed to be insubstantial and did not really provide new ideas. Thus, I suggest a major review to provide more information to help readers to interpret the SEOF methods and their results. Also, for example, how do the results (change in SST amplitudes or variability and teleconnections) make sense physically? Here are some specific comments I've made:

**Major comments**

1. I've found it is a bit difficult to interpret SEOF method intuitively. My understanding is that: conduct EOF analysis over all the ensemble members at each time step, as an analogy to conduct EOF analysis over a time series under a stationary climate. Therefore, each ensemble member here represents each year (under a stationary climate).

However, currently, the majority of climate studies treat ensemble members as different possibilities caused by atmospheric internal variability. The standard deviation of ensemble members is used to evaluate the strength of internal variability (noise), while the ensemble mean is used to present the response to forcings (signal). Thus, in this study, it is confusing when the authors use the std of PC1 to represent the strength of ENSO.

I would suggest to provide more details, leading the readers to easier understand the merit of snapshot framework & SEOF since it is a relatively new method. The current descriptions (in terms of the method) lack of details and difficult to follow (e.g. L41-45; L112-116).

2. The authors kept emphasizing that snapshot framework is better than the traditional temporal statistics method (e.g. L74-77, L231-235). However, the authors did not provide detailed explanations of the pros/cons of both methods, nor did they compare the similarities and discrepancies of the results from the two different methods. Were their results more reasonable (in terms of physics) compared to the ones using temporal statistics method?

I would think that using the time period says from 1900 to the present, it is feasible to compare the results from these two methods with the reanalysis data. By doing so, it would provide a more convincing evidence that the snapshot framework is a more suitable tool.

3. As noted in several recent studies (e.g. Seager et al. 2019 *Nat. Clim. Change*), in most of the state-of-the-art GCMs (including CMIP5), they have an El Nino-like trend in SST over the tropical

Pacific in the warming climate, inconsistent with the observation (that is, increase in the west-east SST gradient -> La Nina-like trend in SST). Does this bias exist in the CESM1? If so, would it affect the fidelity of the research (change in ENSO pattern in the warming climate) here? http://ocp.ldeo.columbia.edu/res/div/ocp/pub/seager/SeagerEtAl2019NC.pdf

4. The correlation analysis in the section 3.2 did not really provide constructively new ideas. The correlation between ENSO SST anomalies and precipitation across the globe has been well-examined and established for decades. And the relationships from this manuscript (e.g. L165-169) are consistent with the previous literature. Therefore, the zero-lag correlation analysis in the current climate here seems to me only demonstrates that CESM1 and the snapshot framework can decently produce ENSO-related SST-precipitation relationships.

   Also, the impacts of ENSO on precipitation (or say teleconnections) cannot be simplified by just examining correlation, especially for boreal summer season. ENSO can be at developing or decaying phases during boreal summer season. The teleconnection patterns and therefore impacts on regional precipitation can be quite different between these two phases. Moreover, El Nino and La Nina have asymmetric characteristics during the decaying phase: an El Nino tends to decay rapidly; while a La Nina tends to decay slowly and even persist into the following winter (e.g. Okumura and Deser 2010). In this context, the teleconnection patterns of El Nino and La Nina are not mirror images, which means, applying correlation analysis on JJAS variables might not be able to reflect the real impacts from El Nino and La Nina on teleconnections. And in this sense, the half-year-lag correlation conducted in this article did not reasonably consider the life-cycle of an ENSO event. https://journals.ametsoc.org/doi/full/10.1175/2010JCLI3592.1

   As the sensitivity of seasonal precipitation over land depends strongly on the configuration and location of teleconnection patterns, I would suggest the authors to include the atmospheric circulation patterns when discussing the change in teleconnection patterns. The circulation patterns could also provide more intuitively physical sense that how the change in tropical SST modulates the large-scale atmospheric circulation and thereby precipitation over the remote area.

5. The title says "On the time evolution of ENSO and its teleconnections in an ensemble view". This title does not explicitly express that the focus of this paper is the change in ENSO under climate change scenario. Instead, "time evolution of ENSO" strongly misleads to the evolution of an ENSO life-cycle (from developing to peak to decaying phases...).

6. In general, as the authors deployed EOF analysis on all the ensemble members at each time step and compared the results from EOF analyses at different time steps. I would suggest that when mentioning variability, change or any analysis used in the article (e.g. linear fit), it would be beneficial to (explicitly) explain that it is over ensemble members or time steps.

7. The authors used "time instant" in this article, but I found this is really confusing. "Instant" gives people a mistaken impression that it means "an infinitesimal space of time". I would suggest that time step is one of the possible alternatives. Or the authors could emphasize that "time instant" used here means seasonal average when "time instant" was mentioned the very first time in the article. Similarly, "instantaneous forcing" (e.g. L48) is also confusing. Is the forcing just turned on for a very short while?

**Minor comments**

**Introduction**
1. L26: However, the model simulations of future ENSO changes diverge widely among climate models. & L36: To avoid the above-mentioned contradiction, in this study we present an ensemble-based analysis

My question is, how could the authors be sure that their method provided the right direction? (Similar to the 2nd major comment).

2. L39-41: Instead of just listing these papers, I would suggest the authors specify some topics that have been examined using large-ensemble when referring these papers.

3. L55: "This approach": what approach? Large ensemble? Or snapshot framework
Seems like it means "large-ensemble" based on the following context.

**Data & Methods**
1. Why use JJAS 4-month average compared to DJF 3-month average?

2. L109: other way around? Consider leading SEOF mode as instantaneous ENSO loading pattern?

3. The authors mentioned Maher et al. 2018 several times (e.g. L105; L142), I am not sure all of them are necessary and provide useful information. As many readers might not read the paper before, so if the authors would like to include it, it would be better to provide more details why the authors need to compare them.

**Results**
1. L131: SST variability? SST anomalies?

2. L150: the explained variance in JJAS is increased (Fig.3b) -> Does this mean ENSO pattern is more favorable in the future? If so, is this consistent with previous studies?

3. Section 3.2: As mentioned in the major comments, JJAS could be during the developing or decaying phases among an ENSO life-cycle, it is important to specify the lag-relation. For example, it is well-known that the Indian Ocean has delayed response (that is during the decaying phase) to an El Nino.

Also, L188-203, it would be much more helpful if the authors could include the change in atmospheric circulations. This would provide more physical sense of how the atmosphere would change given the change in the tropical SST. Just listing the changes in precipitation over some random areas does not really provide essential information for readers to take away.

4. L202: we conclude that a half-year-forward estimate of the precipitation from PC1 data in these regions becomes "more accurate" -> this statement is not accurate. What did the authors mean "more accurate" (in terms of what? forecast?)? How did they assess the accuracy?

**Conclusions**
1. L213-215: Why are the changes in the ENSO pattern in JJAS season larger than in DJF season? Do the authors have any possible physical explanation of this seasonality difference?

**Figures**
In general, the figures are not easy to read. For example, the authors could add titles to each panel. Also, the font size of all the labels are small.

Figure 4 is particularly difficult to read. The coastline contours are not clear. Also, they authors could consider to exclude the high latitude region. The differences between each color interval are not clear enough. It is really difficult to tell the differences. For me, it is just a bunch of red/blue patches. Also, since it is for precipitation, blue (red) might mislead to wetter (drier) condition, so I would suggest to adjust the color bar.

---

## Author Comment (AC1) · 18 Dec 2019

**Reply to Referee #1**

**On the time evolution of ENSO and its teleconnections in an ensemble view – a new perspective**
**by Tímea Haszpra, Mátyás Herein, Tamás Bódai**

*The authors investigate changes in the ENSO phenomenon and its teleconnections using a recently proposed methodology (SEOF). ENSO is the most dominant interannual mode of variability in the climate system: studying changes in its amplitude and its impacts is then relevant to the scientific community. Differently from standard statistical methods in time series analysis, the authors used a new methodology, snapshot empirical orthogonal function (SEOF). SEOF is tailored for ensemble simulations: after a transient time, each member of an ensemble of simulations of a nonlinear dynamical system is believed to cover the distribution of states of the attractor. The authors exploit this to define a new notion of Empirical Orthogonal Function (EOF) along ensemble members, rather than along the time dimension. This methodology allows for the definition of "instantaneous" ENSO patterns. Here the authors consider as "instantaneous" seasonal averages (JJAS and DJF) and proceed in analyzing (a) ENSO pattern and its evolution in time and (b) changes in teleconnections, using the proposed methodology. Changes in the spatial pattern of ENSO (quantified by a regression of the PC of the first SEOF mode on SST) are briefly examined qualitatively and quantitatively by linear fitting the regression maps. Interestingly, the largest changes in patterns are in the JJAS season rather than in DJF. Changes in amplitude of ENSO are also examined. In agreement with previous studies, the authors find an increase in variance of the PC1 and of the Niño 3.4 index. Finally, the authors use the proposed framework to investigate changes in the teleconnection patterns between ENSO and precipitations, using instantaneous and lagged correlations along ensemble members, rather than along the time dimension, and explore the changes of these correlations with time.*

*This is a well written, clear and interesting paper. Also, the application of the SEOF methodology on ENSO is new. I have some minor comments, mainly regarding the methodology.*

Dear Referee #1,

We thank the Referee for the careful reading of our manuscript, and we are glad that the Referee found our manuscript well written and interesting, and that the Referee appreciates the novelty of the SEOF methodology. We are thankful for the Referee's constructive comments on our manuscript. These suggestions have helped improve the presentation and clarity of the paper, and we have incorporated them into the revised manuscript. Here we quote the comments and questions, provide answers and discuss the changes carried out in the text.

*(a) The main strength of this methodology is that it allows to analyze large ensembles in a comprehensive and well defined way. Also this framework offers a route to examine teleconnections, disentangling climate variability and external forcing in a correct way.*

*However, in my opinion, this does not mean that this method is definitively better than traditional time series analysis. Both this methodology and temporal statistics are useful for different reasons. Here some reasons:*

**Response:** We thank the Referee for drawing our attention that a more detailed discussion of the traditional and snapshot frameworks is needed in the manuscript. The Referee's opinion *"this does not mean that this method is definitely better than traditional time series analysis"* led us to clarify the requirements and capabilities of the application of traditional time series analysis tools and those of the snapshot methods. As it is stated in old lines 32–33/in new lines 33–34, stationarity is required for the correct application of the tools of time series analysis, which does not hold in a changing climate. In this case a common practice for separating the forced response of an arbitrary meteorological variable to climate change is to detrend the time series, e.g., by choosing a detrending function (linear, polynomial, exponential, etc.) or applying moving averages over a series of time windows of chosen length. In both cases the choice of the detrending scheme and the length of the time windows brings subjectivity to the analysis, and, furthermore, the results might be misleading (Herein et al. 2017). Even if the process is stationary the choice of a "too short" time period for the analysis may result in detecting false trends (Wunsch 1999, Gershunov et al. 2001, Drótos et al. 2015).

In contrast to this, the snapshot methods, computing statistics at single time instants only from the instantaneous potential states of the climate system, i.e., without the direct effect of previous and future climate states on the statistics of the given time instant, provide a correct way to characterize the plethora of all potential outcomes. These outcomes are compatible with the climate states of the given time instant as determined by an external forcing inducing a climate change, i.e., in the case of processes described by nonstationary statistics. The concept of the snapshot methods with several examples including SEOF analysis was recently reviewed by Tél et al. (2019). In what follows, we detail the changes made to clarify these points in the manuscript.

Drótos, G., Bódai, T., and Tél, T. (2015). Probabilistic concepts in a changing climate: A snapshot attractor picture. Journal of Climate, 28(8), 3275-3288.

Gershunov, A., Schneider, N., and Barnett, T. (2001). Low-frequency modulation of the ENSO–Indian monsoon rainfall relationship: Signal or noise?. Journal of Climate, 14(11), 2486-2492.

Herein, M., Drótos, G., Haszpra, T., Márfy, J., and Tél, T. (2017). The theory of parallel climate realizations as a new framework for teleconnection analysis. Scientific Reports, 7, 44529.

Tél, T., Bódai, T., Drótos, G., Haszpra, T., Herein, M., Kaszás, B., and Vincze, M. (2019): The Theory of Parallel Climate Realizations – A New Framework of Ensemble Methods in a Changing Climate: An Overview. Journal of Statistical Physics (doi:10.1007/s10955-019-02445-7).

Wunsch, C. (1999). The interpretation of short climate records, with comments on the North Atlantic and Southern Oscillations. Bulletin of the American Meteorological Society, 80(2), 245-256.

*- It is true that the choice of the time window is largely subjective. However, ENSO has a quasi-periodicity of 3 to 7 years and its teleconnections can be analyzed a 12-months range (e.g., ENSO leads the Western Indian Ocean with a lead lag of ~3 months). I expect that, given a single member, time windows from 30 to 100 years of data would give robust results. If correlations between two basins, start changing when considering 30 or 100 years it can simply mean that the connections analyzed may not be "stable". This is possible in climate and can be a result of (i) local regime shifts in one of the two basins, causing qualitative changes in local dynamics and so in connections with other basins (see Dekker et al.*

*https://www.earth-syst-dynam.net/9/1243/2018/esd-9-1243-2018.pdf or Klose et al. https://arxiv.org/pdf/1910.12042.pdf) and/or (ii) phenomena of chaotic synchronization between basins (please see this PRL paper from Duane and Tribbia https://pdfs.semanticscholar.org/cd01/9dacfa47fdc2d5b46e8d33dda956fae135b0.pdf). An example of a (possibly) unstable teleconnections is the leading from the Equatorial Atlantic to ENSO. For example, Falasca et al. (https://agupubs.onlinelibrary.wiley.com/doi/epdf/10.1029/2019MS001654) showed that this lead may exist only in certain decades (see Figures 18e, 18f for reanalyses and Figures 19e and 19f for two CESM members), possibly for phenomena of chaotic synchronization. In the case of this specific teleconnection choosing a window of 50 or 100 years would indeed give a different result, but not because of biases in the methodology but because this connection seems to change in time. Also, in the context of the CESM-LE it has been found at different times in different members, suggesting indeed a chaotic synchronization between the two basins.*

**Response:** We agree on the fact that when only single time series are available, such as measurements or single model simulations, one has to use the traditional methods of time series analysis to analyze the phenomena and obtain results by, e.g. in the case of ENSO, assuming a constant EOF loading pattern for a certain time period and studying its oscillation phases and teleconnections by the corresponding PCs. However, during this time period (e.g., within 30 or 100 years) climate may change, and in the case of the CESM-LE climate change indeed manifests in a considerable global surface temperature increase on decadal time scale (Kay et al. 2015). Therefore, neither it can be presupposed that the pattern of the ENSO or the strength of its teleconnections remain constant (as must be assumed for the traditional approach), nor it can be assumed that a single value of regression coefficient or correlation coefficient can faithfully characterize the conditions of several decades. Indeed, as the Referee writes, "*connections analyzed may not be stable"* and traditional results will not be robust.

However, we do not exactly understand what the Referee means by writing "*choosing a window of 50 or 100 years would indeed give a different result, but not because of biases in the methodology but because this connection seems to change in time*". For the very reasons that (i) the connections change in time and (ii) the traditional approach studies connections (and oscillation patterns, etc.) that are treated as constant for the chosen time period (resulting, e.g., in a single value of the correlation coefficient at each grid point for the time period in which the teleconnection is studied), we do think that, at the very least, the traditional result must be biased in the 100-year window compared to those in 50-year windows, and it will also be biased in any of the 50-year windows in a generic case of time dependence. Of course, this bias results from the limitation that there is no mathematically correct and objective way to overcome this problem with single time series.

Kay, J. E., Deser, C., Phillips, A., Mai, A., Hannay, C., Strand, G., ... & Holland, M. (2015). The Community Earth System Model (CESM) large ensemble project: A community resource for studying climate change in the presence of internal climate variability. Bulletin of the American Meteorological Society, 96(8), 1333-1349.

**Change:** To clarify this issue we have added a detailed description about comparing the capabilities of the snapshot and traditional methods and the meaning of their results as the completely new Section 2.3, more than a page long, including the references emphasizing potential impacts of climate change such as shifting of regimes, chaotic synchronization between different regions and the unstable character of teleconnections. We are grateful for the Referee for drawing our attention to these references.

*- Traditional time series analysis (referred in the paper as "temporal statistics") presents lots of desirable tools: (i) different measures of coupling between time series such as linear (e.g., Pearson correlation), nonlinear (e.g., Mutual Information), causal (e.g., PCMCI algorithm) and (ii) robust methodologies to assess statistical significance. This is possible if a large number of data points is analyzed. In the snapshot method, every measure of coupling and every test is constrained by the (very small) number of members of an ensemble. This is a limitation of the methodology since 40 members is still a (very) limited number of data points in the analysis.*

**Response:** We thank the Referee for mentioning that we should draw attention to the limitations of the snapshot methods originating from the small number of ensemble members. We also feel important to repeat here that the desirable tools of traditional time series analysis give robust and unbiased results only if the underlying statistics can be approximated well as stationary. Furthermore, temporal autocorrelation may seriously reduce the effective sample size of the traditional methodology: already with a 3-year autocorrelation, the effective length of a 150-year time series is practically not more than 50 data points.

**Change:** We have added a discussion on the limitation due to small ensemble sizes to new Section 2.3 immediately after discussing the temporal and snapshot methods according to the previous Referee question.

*- More importantly, all results of the snapshot methodology, live in model-land (see http://www.economics-ejournal.org/economics/discussionpapers/2019-23/file). In fact, in reality we have only access to one climate and we have no access to an ensemble. Therefore, the results that can be obtained using this methodology, while interesting, are always going to be constrained to the chosen climate model and its biases. These points should be briefly discuss. Advantages and disadvantages of both the snapshot methodology and traditional time series analysis should be made clear. My view is that they are both useful and can complement each other, and not that one is definitive better than the other.*

**Response:** We thank the Referee for drawing our attention to the importance of emphasizing even more in the paper that snapshot methods can only be applied when an ensemble of climate realizations is available. Most often, such an ensemble is produced by a climate model, however, they may also be accessible in experiments. For example, they proved to be useful in laboratory experiments aiming to study the effect of climate change on mid-latitude atmospheric circulation (Vincze et al. 2017). Obviously, the obtained results are constrained by the climate model or the capability of the experimental setup.

Vincze, M., Borcia, I. D., and Harlander, U. (2017). Temperature fluctuations in a changing climate: an ensemble-based experimental approach. Scientific Reports, 7(1), 254.

**Change:** We have added the discussion in this Response to the last paragraph of Section 2.3 as well.

*(b) Figure 2. Are the trends of the regression maps really linear? Was this checked? I would have expected to be linear in a time range of ~30 years but not necessarily from 1950 and 2100. Can you please check two random time series in the ENSO region and in the Horse*

*Shoe Pattern (the region with strong negative linear trend) in Fig. 2a and see the shape of the trend?*

**Response:** Although it was not checked, but based on our experience regarding the shape of the time series of the ENSO strength, explained variance of the first SEOF mode and Niño3 amplitude in Fig. 3 we expected that the changes in the regression maps can be approximated by linear trends as well. However, motivated by the Referee's remark, we have checked the time series of two grid points in the ENSO region and two in the Horse Shoe Pattern in Fig. 2, and have found that they can indeed be well approximated by linear trends.

**Change:** We have added a sentence in old line 134/in new line 228–231 regarding the shape of the trends in the regression maps, and included Fig. S1 to the Supplement in order to provide some examples on the time series of the regression coefficients at different geographical locations.

*(c) Figure 3. Panels (b) and (e). It is interesting to see that while the explained variance of PC1 in DJF is relatively constant, this is not true for the season JJAS. In Figure 3b the explained variance of the PC1 experiences a steady increase from ~45% to ~60%. It could be interesting to analyze the second mode of the SEOF and see how it is changing. If this analysis would help in better understanding (or at least suggests an explanation) the increase in variance of the first mode I would recommend to add the analysis of the second mode in the appendix.*

**Response:** Motivated by the Referee's question, we have checked the change not only in the second mode of the SEOF analysis but also in higher modes, up to the fifth one. The increasing values in the explained variance of the first SEOF mode are found to be compensated by the generally slightly decreasing trends appearing in the explained variance of other modes. In JJAS, the second mode contributes the most to the decrease, by 2.4%, while in DJF, for which Fig. 3.e shows a less pronounced increase for the first mode, the explained variance of the second mode is approximately constant, and the compensating decrease appears in the explained variance of the higher-order modes.

**Change:** These new findings are added to old line 152/to new lines 262–267, and new Fig. S2 is added to the Supplement to demonstrate the change in the explained variance of modes #2–#5.

*(d) Figure 5. Second line of the caption. Correct: DJF PRECT (b → DJF PRECT (b)*

**Change:** We thank the Referee, we have corrected it.

We hope that these amendments and added discussion address the Referee's concerns.

Yours sincerely,

Tímea Haszpra, Mátyás Herein, Tamás Bódai

---

## Author Comment (AC2) · 18 Dec 2019

**Reply to Short Comment #1 by Sebastian Milinski**

**On the time evolution of ENSO and its teleconnections in an ensemble view – a new perspective**
**by Tímea Haszpra, Mátyás Herein, Tamás Bódai**

*This manuscript provides some interesting ideas by building on previous work that used the ensemble dimension in a large ensemble to describe forced changes in the statistics of the climate system. In particular changes in teleconnections, here characterised by the correlation coefficient in the ensemble dimension, may provide some new insights.*

*I was wondering why you decided to focus on the first EOF to characterise ENSO variability. Takahashi et al. (2011) argue that both EOF1 and EOF2 should be used to characterise ENSO. Did you test if there are changes in the second EOF? Changes in EOF 2 might also project on the Nino3 region and could theoretically even have an opposing effect compared to the changes in EOF 1 discussed in this manuscript.*

**Response:** Thank you for your comment. As you argue, the EOF2 mode can certainly have a role in the characterization of ENSO variability. Our main intention was to show the applicability and advantages of the SEOF analysis on a simple and easy-to-follow example. Therefore, we investigated the "conventional" EOF pattern of ENSO, i.e., the one associated with the EOF1 mode, following, e.g., Diaz et al. (2001), as mentioned in line 118–121 in the original manuscript. Other studies, e.g., Ashok et al. (2007) (also cited in these lines) confirm that this quantity and the derived PC1 are strongly correlated with the SST variability in the Niño3 region and with the Niño3 index, respectively, while this does not hold for the EOF2 mode and PC2. /Quote from Ashok et al. (2007): "The correlation between PC1 and NINO3 index is very high, and amounts to 0.98, which proves that EOF1 represents the conventional El Niño well. On the other hand, the correlation between PC2 and NINO3 index is very low (−0.09)."/

Nevertheless, motivated by Referee #1, we carried out a new analysis on studying the changes in the explained variance of the EOF2. Fig. S2 in the Supplement illustrates that there is a slight decrease in JJAS, and no significant change in DJF.

Ashok, K., Behera, S. K., Rao, S. A., Weng, H., and Yamagata, T. (2007). El Niño Modoki and its possible teleconnection. Journal of Geophysical Research: Oceans, 112(C11).

Diaz, H. F., Hoerling, M. P., and Eischeid, J. K. (2001). ENSO variability, teleconnections and climate change. International Journal of Climatology, 21(15), 1845-1862.

**Change:** In order to better support our choice on EOF1, we have added the value of the correlation coefficient found by Ashok et al. (2017) between the PC1 and the Niño3 index to the sentence in old lines 118–121/in new lines 139–141, and we have emphasized that despite the existence of more complex indices for the characterization of ENSO (such as the ones in Takahashi et al. (2011)) we choose PC1 to provide a simple and easy-to-follow example on illustrating the applicability and advantages of the SEOF analysis.

Takahashi, K., Montecinos, A., Goubanova, K., and Dewitte, B. (2011). ENSO regimes: Reinterpreting the canonical and Modoki El Niño. Geophysical research letters, 38(10).

*How much does the sampling uncertainty affect the detected changes? I.e. how much of the difference in variance between two years can be attributed to the forcing change and how much of the difference is due to the limited ensemble size? Note that we concluded in Maher et al. (2018) that 30-40 ensemble members are sufficient to quantify ENSO variability when analysing ENSO variability over time periods of 10-50 years. Arguably, 10 years and 30 members might not even be sufficient, depending on the acceptable error (figure 4 in Maher et al. 2018). Since you are using indvidual years, it could be possible that more than 30-40 members are required. Based on this, I would expect to see large sampling uncertainty in the correlation coefficients. It might be beneficial to show the time series for the correlation coefficients for some selected regions to demonstrate that the discussed changes are larger than the sampling uncertainty.*

**Response:** We note that already in Fig. 3 of the previous version of the manuscript trends in ENSO strength, explained variance of the first SEOF mode and Niño3 amplitude proved to be detectable on a traditionally computed 95% significance level using the CESM-LE. It means that 30-40 members over the studied 150 years with the prescribed RCP8.5 forcing proved to be sufficient to detect the changes in the time series despite the considerable magnitude of the fluctuations due to the sampling uncertainty deriving from the number of ensemble members.

**Change:** Motivated by this question, we indicate in all of the map figures of the new version of the manuscript the geographical locations where correlations or detected trends are significant at the traditionally computed 95% level. Based on these data, we may safely state that the number of ensemble members in this study is sufficient to characterize the strength of the teleconnections reasonably well and to detect the changes during the investigated 150 years.

As the statement regarding the number of ensemble members was the result of a misunderstanding in this context in line 88 of the original manuscript/in line 101 of the revised manuscript, we deleted this part of the sentence in the new version.

*Separating amplitude and pattern changes: In figure 1, you standardised the PC1. Thus both pattern and amplitude changes, if they occur, can be seen in the regression maps. Did you use the same approach for the analysis in figure 3? An alternative approach to separate pattern and amplitude changes would be to normalise the pattern. Amplitude changes can then be seen in the PC, whereas pattern changes can be seen by comparing the regression maps for different states of the climate system. This is the approach we used in Maher et al. (2018).*

**Response:** We would like to note that, as it can be seen from the values and the unit, in Fig. 3 PC1 is not standardized, rather it includes the mentioned separated amplitude of the oscillation.

Besides studying the separated amplitude of the ENSO phenomenon, we felt more intuitive and meaningful to present regression maps and analyze the changes in them, as these maps show the typical value of the amplitude of the fluctuations directly related to the given EOF mode of variability at each grid point, which in the case of EOF1 has the strongest relationship with ENSO. The changes in the regression maps are then easy to interpret: they show the changes in these fluctuation amplitudes, i.e., changes in the typical SST anomalies

bound to the given mode at each grid point, and potential shifting in the pattern during climate change as well.

In contrast to this, from normalized patterns or EOF loading patterns the value of the typical SST anomalies bound to the given mode cannot be seen, just the fact that at some locations they are somewhat greater than at other ones or are in the opposite phase at a given time instant. The changes in these normalized patterns show the changes in the relative importance of different regions over time from the point of view of the given mode. Motivated by the comment, we carried out the analysis of the separated pattern as well, using the raw SEOF loading patterns (by which we mean the normalized eigenvectors associated with the largest eigenvalue of the covariance matrix of the SST anomaly fields).

**Change:** As we feel more intuitive to illustrate the changes in the ENSO pattern by regression maps, we keep them in the new version of the manuscript. However, as a supplement to the regression maps, we have added Fig. S3 to the Supplement containing the maps of the first SEOF modes (SEOF loading patterns) for the years in Fig. 1. We have also carried out the linear regression analysis at each grid point, analogously to Fig. 2, to track the changes in the separated pattern. These are displayed in Fig. S4. The interpretation of these results are added in old line 142/in new line 238–250.

---

## Author Comment (AC3) · 18 Dec 2019

**Reply to Referee #2**

**On the time evolution of ENSO and its teleconnections in an ensemble view – a new perspective by Tímea Haszpra, Mátyás Herein, Tamás Bódai**

In this paper, the authors aimed to examine changes in ENSO SSTA patterns and teleconnections under climate change, using a recently developed ensemble-based method (SEOF). SEOF was applied on all the ensemble members from CESM1 at each time step, avoiding using statistical mean or standard deviation under a non-stationary climate.

This ensemble-based method provides a very interesting perspective to study change in the warming climate. This manuscript showed how to apply this method on the change in ENSO patterns. However, I do feel this manuscript lacks details in terms of the physical interpretation of the method (SEOF) and results, which makes the article quite difficult to follow. Also, the content of change in teleconnections which only used correlation to analyze (Section3.2) seemed to be insubstantial and did not really provide new ideas. Thus, I suggest a major review to provide more information to help readers to interpret the SEOF methods and their results. Also, for example, how do the results (change in SST amplitudes or variability and teleconnections) make sense physically? Here are some specific comments I've made:

**Dear Referee #2,**

We thank the Referee for the careful reading of our manuscript, and we are glad that the Referee thinks that the newly developed SEOF method provides a very interesting perspective to study the effects of climate change. We appreciate the Referee's constructive comments on our manuscript. These suggestions have helped improve the presentation and clarity of the paper, and we have incorporated them into the revised manuscript.

As we indicate in our responses, the physical and mathematical interpretation of the SEOF analysis is now elaborated in a new section (Section 2.3). We emphasize that Sec. 3.2 for the correlation analysis characterizing the strength of teleconnections and their change was not intended to provide new ideas, but rather to illustrate how to apply the SEOF-derived PC1s for teleconnection analysis within the snapshot framework, and to reveal the capability of the CESM-LE and the SEOF analysis regarding the ENSO teleconnections compared to the observations. Furthermore, this section is also devoted to study the forced changes in the strength of teleconnections, which has not been done so far for ENSO teleconnections in the snapshot framework to the best of our knowledge. Here we quote the specific comments and questions, provide answers, and discuss the changes carried out in the text.

**Major comments**

1. I've found it is a bit difficult to interpret SEOF method intuitively. My understanding is that: conduct EOF analysis over all the ensemble members at each time step, as an analogy to

conduct EOF analysis over a time series under a stationary climate. Therefore, each ensemble member here represents each year (under a stationary climate).

However, currently, the majority of climate studies treat ensemble members as different possibilities caused by atmospheric internal variability. The standard deviation of ensemble members is used to evaluate the strength of internal variability (noise), while the ensemble mean is used to present the response to forcings (signal). Thus, in this study, it is confusing when the authors use the std of PC1 to represent the strength of ENSO.

I would suggest to provide more details, leading the readers to easier understand the merit of snapshot framework & SEOF since it is a relatively new method. The current descriptions (in terms of the method) lack of details and difficult to follow (e.g. L41-45; L112-116).

**Response:** Although the ensemble members at a given time step can be considered as the analogue of the years under a stationary climate, we would formulate the first paragraph of this comment in a slightly different way: a regression map (or the loading pattern) derived from EOF analysis in time series analysis represents the spatial pattern of a standing oscillation that characterizes the temporal variability in the SST over the chosen time interval: as it is stated in old line 115/in new line 133, the regression maps show typical amplitudes of the SST anomalies at each grid point (with respect to the mean state of the climate represented by the temporal mean of the SST fields). Analogously, the ensemble-based regression map of the SEOF analysis, obtained at a chosen time instant, represents an oscillation which characterizes the potential variability in SST across the ensemble (with respect to the instantaneous mean state represented by the ensemble mean of the SST fields), i.e., it describes a kind of interval variability of the climate system specific to the given time instant, similarly to the ensemble standard deviation mentioned by the Referee. In fact, the temporal fluctuations that constitute internal variability in a stationary climate is nothing else but the manifestation of all the different possibilities permitted by the chaotic dynamics of the system. The ensemble spread is the instantaneous analogue of these fluctuations. For more details, see Drótos et al. (2015, 2017) and Tél et al. (2019).

Regarding the second paragraph of the comment, in the traditional EOF analysis using single time series, the (temporal) standard deviation of the PC1 is a common practice to represent the strength of an oscillation that is derived by EOF analysis (for ENSO, see, e.g., Monahan and Dai 2004, Maher et al. 2018). Therefore, in this study, we use its ensemble-based counterpart to represent the strength of the ENSO at a given time instant.

Drótos, G., Bódai, T., and Tél, T. (2015). Probabilistic concepts in a changing climate: A snapshot attractor picture. Journal of Climate, 28(8), 3275-3288.

Drótos, G., Bódai, T., and Tél, T. (2017). On the importance of the convergence to climate attractors. The European Physical Journal Special Topics, 226(9), 2031-2038.

Maher, N., Matei, D., Milinski, S., and Marotzke, J.: ENSO change in climate projections: forced response or internal variability?, Geophysical Research Letters, 45, 11–390, 2018.

Monahan, A. H., and Dai, A.: The spatial and temporal structure of ENSO nonlinearity. Journal of Climate, 17(15), 3026-3036, 2004.

Tél, T., Bódai, T., Drótos, G., Haszpra, T., Herein, M., Kaszás, B., and Vincze, M. (2019): The Theory of Parallel Climate Realizations – A New Framework of Ensemble Methods in a Changing Climate: An Overview. Journal of Statistical Physics (doi:10.1007/s10955-019-02445-7).

**Change:** For clarity, we have added the term "ensemble" before the "standard deviation of the PCs" at each occurence. Furthermore, for clarity, we have added to old line 116/new lines 134–136 that "The instantaneous strength of ENSO is computed as the ensemble

standard deviation of the PC1s of the given time instant as the snapshot counterpart of the temporal standard deviation of the PC1 used as a common practice to represent the strength of an oscillation in traditional EOF analysis (e.g., Monahan and Dai, 2004; Maher et al., 2018)."

For an easier understanding of the merit of snapshot framework and SEOF analysis, a completely new Section, 2.3 is devoted to a detailed presentation of the capabilities of the snapshot methods compared to the traditional ones and to interpret the meaning of their results. It includes a discussion recalling that temporal statistics are meant to be evaluated for stationary time series only, whereas the snapshot methods can handle nonstationary processes as well.

2. The authors kept emphasizing that snapshot framework is better than the traditional temporal statistics method (e.g. L74-77, L231-235). However, the authors did not provide detailed explanations of the pros/cons of both methods, nor did they compare the similarities and discrepancies of the results from the two different methods. Were their results more reasonable (in terms of physics) compared to the ones using temporal statistics method?

I would think that using the time period says from 1900 to the present, it is feasible to compare the results from these two methods with the reanalysis data. By doing so, it would provide a more convincing evidence that the snapshot framework is a more suitable tool.

**Change:** In new Section 2.3 detailed explanations of the snapshot and temporal methods, and their pros/cons are presented. In the traditional temporal approach connections (and oscillation patterns, etc.) are treated as constant for a chosen time period (resulting, e.g., in a single value of correlation coefficient at each grid point for the time period for studying teleconnections). In contrast to this snapshot methods use only the information of the potential outcomes compatible with the climate states of the given time instant, without the direct impact of previous or future climate states on the value of the statistics. Therefore, the results obtained from snapshot methods are more reasonable in terms of physics.

**Response:** As Section 2.3 mentions, a comparison of results derived from snapshot methods and time series analysis can be found in Herein et al. (2017) and Bódai et al. (2019) on the example of the North Atlantic Oscillation teleconnections using a station-based NAO index and the ENSO phenomenon using the Niño3 és SOI indices. The single time series results were shown to be strongly different from the snapshot ones. These papers also illustrate by numerical examples that the choice of the time window may have a considerable effect on the statistical measures in the traditional approach, while this is not a problem when using the snapshot framework.

Bódai, T., Drótos, G., Herein, M., Lunkeit, F., and Lucarini, V. (2019). The forced response of the El Niño–Southern Oscillation-Indian monsoon teleconnection in ensembles of Earth System Models. Journal of Climate, https://doi.org/10.1175/JCLI-D-19-0341.1.

Herein, M., Drótos, G., Haszpra, T., Márfy, J., and Tél, T. (2017). The theory of parallel climate realizations as a new framework for teleconnection analysis. Scientific Reports, 7, 44529.

3. As noted in several recent studies (e.g. Seager et al. 2019 Nat. Clim. Change), in most of the state-of-the-art GCMs (including CMIP5), they have an El Nino-like trend in SST over the tropical Pacific in the warming climate, inconsistent with the observation (that is, increase in the west-east SST gradient -> La Nina-like trend in SST). Does this bias exist in the

CESM1? If so, would it affect the fidelity of the research (change in ENSO pattern in the warming climate) here?

**Response:** We thank the Referee for drawing our attention to the fact that the so-called Niño3.4 SST trend might affect the performance of the CESM-LE in capturing the ENSO pattern and its changes. The study suggested by the Referee (Seager et al. 2019) also includes results obtained for CESM-LE called the "National Center for Atmospheric Research (NCAR) Large Ensemble (LENS)" in the study (for name convention, see subsection "CMIP5 models and NCAR LENS" in section "Methods").

According to the paper, similarly to CMIP5 models, the ensemble mean of LENS also shows a moderate Niño3.4 trend over 60 years inconsistent with HadISST and NCEP/NCAR reanalysis, however, this trend proves to be smaller than the CMIP5 multimodel mean for the studied time interval of end years 2008–2017, and some of the ensemble members approach well the values derived from reanalysis. Furthermore, the ECMWF/ORAS4 reanalysis trend values are quite close to LENS ensemble mean for end years of 2008–2009.

The Niño3.4 trend may have an effect on the strength of and change in the teleconnections, however, since Section 3.2 proves that the results from CESM-LE obtained by SEOF analysis are roughly consistent with the observed teleconnections and the CESM-LE performs relatively well according to Seager et al. (2019), we expect that it does not influence much the strength and changes of the connections found in this study.

Seager, R., Cane, M., Henderson, N., Lee, D. E., Abernathey, R., and Zhang, H.: Strengthening tropical Pacific zonal sea surface temperature gradient consistent with rising greenhouse gases. *Nature Climate Change*, *9*(7), 517, 2019.

**Change:** In order to address this question, a discussion about the deviation in the SST trends in the reanalysis data and in CESM-LE has been added to the Conclusions in old line 231/in new line 354–365.

4. The correlation analysis in the section 3.2 did not really provide constructively new ideas. The correlation between ENSO SST anomalies and precipitation across the globe has been well-examined and established for decades. And the relationships from this manuscript (e.g. L165-169) are consistent with the previous literature. Therefore, the zero-lag correlation analysis in the current climate here seems to me only demonstrates that CESM1 and the snapshot framework can decently produce ENSO-related SST-precipitation relationships.

Also, the impacts of ENSO on precipitation (or say teleconnections) cannot be simplified by just examining correlation, especially for boreal summer season. ENSO can be at developing or decaying phases during boreal summer season. The teleconnection patterns and therefore impacts on regional precipitation can be quite different between these two phases. Moreover, El Nino and La Nina have asymmetric characteristics during the decaying phase: an El Nino tends to decay rapidly; while a La Nina tends to decay slowly and even persist into the following winter (e.g. Okumura and Deser 2010). In this context, the teleconnection patterns of El Nino and La Nina are not mirror images, which means, applying correlation analysis on JJAS variables might not be able to reflect the real impacts from El Nino and La Nina on teleconnections. And in this sense, the half-year-lag correlation conducted in this article did not reasonably consider the lifecycle of an ENSO event. https://journals.ametsoc.org/doi/full/10.1175/2010JCLI3592.1

As the sensitivity of seasonal precipitation over land depends strongly on the configuration and location of teleconnection patterns, I would suggest the authors to include the atmospheric circulation patterns when discussing the change in teleconnection patterns. The circulation patterns could also provide more intuitively physical sense that how the change in tropical SST modulates the large-scale atmospheric circulation and thereby precipitation over the remote area.

**Response:** As the Referee writes, Sec. 3.2 was not intended to provide new ideas, but rather to illustrate (1) how to apply the SEOF-derived PC1s for teleconnection analysis within the snapshot framework, and (2) to reveal the capability of the CESM-LE and the SEOF analysis regarding the ENSO teleconnections compared to the observations (similarly to what is done in Section 3.1 revealing the ENSO patterns and amplitudes in CESM-LE using SEOF). We feel that it is satisfying that these results are roughly consistent with the previous literature analyzing observation-based data. Furthermore, this section is also devoted to study the forced changes in the strength of teleconnections, which has not been done so far for ENSO teleconnections in the snapshot framework to the best of our knowledge.

Regardless of the asymmetric characteristics of the decaying or developing phase of El Niño and La Niña, PC1 characterizes the instantaneous phase of the ENSO. Although the connection with the corresponding PRECT, of course, can be quite different in different phases and could thus be described by more sophisticated techniques as well, a kind of leading-order characterization is also viable in terms of the correlation coefficient, and the strength of connection can be defined in this way. We note that this characterization is also used, e.g., by Krishna Kumar et al. (1999) and Ramu et al. (2018) between the Niño3/Niño3.4 index and precipitation, too. We agree with the Referee, however, that when one wishes to maximize predictability or the explanatory power, then further "dimensions" of the problem should be considered, which we feel could be a topic of further research.

Regarding the last paragraph of the question on atmospheric circulation patterns, we intended to devote this study to illustrate the applicability of the snapshot framework to the ENSO phenomenon, and to present alterations in the expected conditions by 2100 based on CESM-LE. We think that an overall and profound investigation of the reasons behind the observed changes would be an enormous project and could also be a topic of future research.

Krishna Kumar, K., Rajagopalan, B., and Cane, M. A.: On the weakening relationship between the Indian monsoon and ENSO. Science, 284(5423), 2156-2159, 1999.

Ramu, D.A., Chowdary, J.S., Ramakrishna, S.S.V.S., and Kumar, O. S. R. U. B.: Diversity in the representation of large-scale circulation associated with ENSO-Indian summer monsoon teleconnections in CMIP5 models. Theor. Appl. Climatol. 132. 1-2, 465–478, 2018.

5. The title says "On the time evolution of ENSO and its teleconnections in an ensemble view". This title does not explicitly express that the focus of this paper is the change in ENSO under climate change scenario. Instead, "time evolution of ENSO" strongly misleads to the evolution of an ENSO life-cycle (from developing to peak to decaying phases...).

**Change:** Motivated by the Referee's suggestion, we have changed the title to "Investigating ENSO and its teleconnections under climate change in an ensemble view – a new perspective".

6. In general, as the authors deployed EOF analysis on all the ensemble members at each time step and compared the results from EOF analyses at different time steps. I would suggest that when mentioning variability, change or any analysis used in the article (e.g. linear fit), it would be beneficial to (explicitly) explain that it is over ensemble members or time steps.

**Change:** Thank you for your suggestion, we have added "over ensemble"/"ensemble-based" or "over time" at each of these occurrences.

7. The authors used "time instant" in this article, but I found this is really confusing. "Instant" gives people a mistaken impression that it means "an infinitesimal space of time". I would suggest that time step is one of the possible alternatives. Or the authors could emphasize that "time instant" used here means seasonal average when "time instant" was mentioned the very first time in the article. Similarly, "instantaneous forcing" (e.g. L48) is also confusing. Is the forcing just turned on for a very short while?

**Response:** Thank you for drawing our attention to this misunderstable term.

**Change:** We have brought forward the relevant sentence from old line 106 to the first occurence of the term "time instant" in old line 37/in new line 39. This sentence has also been rephrased from "On a ``time instant" we mean seasonal average: note that a season can be considered short, but the snapshot framework is also applicable for quantities evaluated over time intervals (Drótos et al., 2015)" to "We note that a ``time instant" can also mean time averages over certain periods, because the snapshot framework is also applicable for quantities evaluated over time intervals (Drótos et al., 2015)".

The term "instantaneous forcing" has been changed to "the external forcing history up to that time".

**Minor comments**

**Introduction**

1. L26: However, the model simulations of future ENSO changes diverge widely among climate models. & L36: To avoid the above-mentioned contradiction, in this study we present an ensemble-based analysis. My question is, how could the authors be sure that their method provided the right direction? (Similar to the 2nd major comment).

**Response:** As detailed in the answer for the 2nd major comment, the new Section 2.3 shows why ensemble-based snapshot methods are correct in terms of the physics characterizing the plethora of all potential outcomes compatible with the instantaneous climate states. The text "above-mentioned contradiction" referred to the paragraph just above this sentence, so it is about the disadvantages of using temporal statistics and not to address the problem of the large divergence of ENSO changes among climate models two paragraphs earlier.

Change: For clarity, "contradiction" is changed to "discrepancy of temporal methods".

2. L39-41: Instead of just listing these papers, I would suggest the authors specify some topics that have been examined using large-ensemble when referring these papers.

**Change:** We have detailed the topics of these references in new lines 41–52. that have been examined using large-ensembles.

3. L55: "This approach": what approach? Large ensemble? Or snapshot framework. Seems like it means "large-ensemble" based on the following context.

**Change:** Thank you for this remark, for clarity, we have changed "This approach" to "The snapshot framework, which can be applied numerically to large ensembles,".

**Data & Methods**

**1. Why use JJAS 4-month average compared to DJF 3-month average?**

**Response:** As ENSO has its maximum around boreal winter, which is traditionally defined as DJF, we analyze the DJF ENSO pattern and ENSO teleconnections in the paper. In order to investigate the possibility of predicting precipitation half a year in advance based on PC1, we calculate lagged correlations beyond instantaneous ones. The relationship between ENSO and the South Asian monsoon is believed to be one of the most important teleconnection phenomena and is traditionally investigated using JJAS (see, e.g. Krishna Kumar et al. (1999), Ashok et al. (2007), Srivastava et al. (2019)), and West Africa also receives the major proportion of its annual rainfall in JJAS (Srivastava et al. (2019)), therefore, we chose JJAS. The choice of the 3-month long DJF season combined with the 4-month long JJAS season is also used by Wu et al. (2012) for studying the ENSO influences on Indian summer monsoon.

Ashok, K., Behera, S. K., Rao, S. A., Weng, H., and Yamagata, T. (2007). El Niño Modoki and its possible teleconnection. Journal of Geophysical Research: Oceans, 112(C11).

Krishna Kumar, K., Rajagopalan, B., and Cane, M. A. (1999). On the weakening relationship between the Indian monsoon and ENSO. *Science*, *284*(5423), 2156-2159.

Srivastava, G., Chakraborty, A., and Nanjundiah, R. S. (2019). Multidecadal see-saw of the impact of ENSO on Indian and West African summer monsoon rainfall. Climate dynamics, 52(11), 6633-6649.

Wu, R., Chen, J., and Chen, W. (2012). Different types of ENSO influences on the Indian summer monsoon variability. Journal of Climate, 25(3), 903-920.

**Change:** To clarify the choice of DJF and JJAS, we have added a short description about the above reasons in old line 124/in new lines 148–154.

2. L109: other way around? Consider leading SEOF mode as instantaneous ENSO loading pattern?

**Change:** Thank you for your remark. We have rephrased the sentence to "We consider the instantaneous **ensemble-based** leading SEOF mode (by which we mean the normalized eigenvectors associated with the largest eigenvalue of the covariance matrix of the SST anomaly fields) as the ENSO loading pattern, ..."

3. The authors mentioned Maher et al. 2018 several times (e.g. L105; L142), I am not sure all of them are necessary and provide useful information. As many readers might not read the paper before, so if the authors would like to include it, it would be better to provide more details why the authors need to compare them.

**Response:** We believe that emphasizing the comparison is important because they applied a similar but somewhat different technique, called EOF-E to analyze the ENSO phenomenon, as it is stated in old line 65/in new line 77. It is straightforward and natural that the results of the two methods are worth comparing, especially because these alternative propositions are very fresh.

**Change:** Where we first compare our results to those of Maher et al. (2018), in old line 142/in new line 237, we now indicate that the reason for comparison is the similarity of the EOF-E and SEOF analyses. (This has also been mentioned in the Introduction, already in the original manuscript.)

**Results**

**1. L131: SST variability? SST anomalies?**

**Change:** We have rephrased the sentence to "... the typical amplitudes of the SST anomaly values across the ensemble members at the Equatorial Pacific are somewhat larger in DJF...".

2. L150: the explained variance in JJAS is increased (Fig.3b) -> Does this mean ENSO pattern is more favorable in the future? If so, is this consistent with previous studies?

**Response:** The larger values of the explained variance mean that by 2100 the oscillation associated with the first mode is going to be responsible for a much larger fraction of the variability in the SST fields. We could not find any previous study about this finding.

**Change:** We have added this sentence in old line 152/in new line 260–267 to better explain the meaning of our results. This also includes a description of the changes in the higher modes.

3. Section 3.2: As mentioned in the major comments, JJAS could be during the developing or decaying phases among an ENSO life-cycle, it is important to specify the lag-relation. For example, it is well-known that the Indian Ocean has delayed response (that is during the decaying phase) to an El Nino.

Also, L188-203, it would be much more helpful if the authors could include the change in atmospheric circulations. This would provide more physical sense of how the atmosphere would change given the change in the tropical SST. Just listing the changes in precipitation over some random areas does not really provide essential information for readers to take away.

**Response:** In addition to our response to these two suggestions above, we would like to note that we did mean to give some indication that the "life cycle of the processes" involved does have a significance. This is why we considered lagged correlations, beside our intention to indicate predictability. However, we do suspect that the analysis framework provided by (lagged) correlations has its limits, such as – what the Referee pointed out – it should matter that it is a developing or decay phase of the El Niño (not just the value of a Niño index or PC1), or, the time lag could be varied in a range around zero to better resolve the phenomenon. However, such a methodology implies a large breadth of visuals to analyse, which is likely not a way to go in order to have a good understanding. Nevertheless, correlations are already very useful to evaluate, and doing this in an ensemble-based framework in order to detect the forced response of teleconnections is a very recent proposition, and, therefore, should not be considered trivial.

4. L202: we conclude that a half-year-forward estimate of the precipitation from PC1 data in these regions becomes "more accurate" -> this statement is not accurate. What did the authors mean "more accurate" (in terms of what? forecast?)? How did they assess the accuracy?

**Response:** The cited line is "The lagged correlations for DJF PC1 and JJAS PRECT (Fig. 5.c) are found to increase considerably near the eastern coast of Africa, in the Niño3 and Niño4 regions and around the Caribbean Islands. Thus, we conclude that a half-year-forward estimate of the precipitation from PC1 data in these regions becomes more accurate."

It means that the correlation coefficient is larger, i.e., the relationship between them is stronger, i.e., there is "more chance" to predict the PRECT amount based on PC1 data than in a scenario of weaker correlations.

Change: For clarity, we added to the sentence that "Thus, based on the larger value of the correlation coefficient implying a stronger relationship, we conclude ..."

**Conclusions**

1. L213-215: Why are the changes in the ENSO pattern in JJAS season larger than in DJF season? Do the authors have any possible physical explanation of this seasonality difference?

**Response:** In general, a larger change in the different quantities is found for JJAS than for DJF. While at the beginning of the JJAS season the ENSO cycle is generally just switching phase in the CESM-LE (Wieners et al. 2019), DJF can be considered to be the "main" ENSO season with the largest SST anomalies. The smaller changes in the DJF quantities may be explained by the conjecture that, calculated for the main ENSO season, the DJF characteristics may be more robust and, thus, undergo weaker alterations during the investigated 150 years than the JJAS ones, which are calculated around the phase change of the cycle. A more thorough investigation of this question could be a topic of future research.

Wieners, C. E., Dijkstra, H. A., and de Ruijter, W. P. (2019). The interaction between the Western Indian Ocean and ENSO in CESM. Climate Dynamics, 52(9-10), 5153-5172.

**Change:** The above explanation has been added to the section Conclusion in new lines 337–342.

**Figures**

*In general, the figures are not easy to read. For example, the authors could add titles to each panel. Also, the font size of all the labels are small.*

Figure 4 is particularly difficult to read. The coastline contours are not clear. Also, they authors could consider to exclude the high latitude region. The differences between each color interval are not clear enough. It is really difficult to tell the differences. For me, it is just a bunch of red/blue patches. Also, since it is for precipitation, blue (red) might mislead to wetter (drier) condition, so I would suggest to adjust the color bar.

**Response:** Motivated by the Referee's remark, we have improved the figures.

**Change:** We have carried out the following changes on the figures:

- titles include the months, and in the case of correlation maps also the studied quantities have been added to each panel,

- the font size of the labels and axes has been enlarged,

- the color of the coastlines in each figure has been changed from gray to black and their line width is thicker now,

- for better visibility, the colorbar of the figures in Figs. 1-2 is changed for another one with more different red and blue shades,

- to better represent drier and wetter conditions, the colorbar of Figs. 4-5 is changed to a brown-green one,

- motivated by the question of SC1 about the sampling uncertainty vs. detectable changes, in all of the concerned figures the geographical locations where correlations or detected trends are significant at the traditionally computed 95% level are indicated.

We hope that these amendments and added discussion address the Referee's concerns.

Yours sincerely,

Tímea Haszpra, Mátyás Herein, Tamás Bódai